# Distributed Conformal Prediction via Message Passing

**Haifeng Wen\*** [1]  **Hong Xing\*** [1,2]  **Osvaldo Simeone** [3]

## Abstract

Post-hoc calibration of pre-trained models is critical for ensuring reliable inference, especially in safety-critical domains such as healthcare. Conformal Prediction (CP) offers a robust post-hoc calibration framework, providing distribution-free statistical coverage guarantees for prediction sets by leveraging held-out datasets. In this work, we address a decentralized setting where each device has limited calibration data and can communicate only with its neighbors over an arbitrary graph topology. We propose two message-passing-based approaches for achieving reliable inference via CP: quantile-based distributed conformal prediction (Q-DCP) and histogram-based distributed conformal prediction (H-DCP). Q-DCP employs distributed quantile regression enhanced with tailored smoothing and regularization terms to accelerate convergence, while H-DCP uses a consensus-based histogram estimation approach. Through extensive experiments, we investigate the trade-offs between hyperparameter tuning requirements, communication overhead, coverage guarantees, and prediction set sizes across different network topologies. The code of our work is released on: `https://github.com/HaifengWen/Distributed-Conformal-Prediction`.

## 1. Introduction

### 1.1. Context and Motivation

The post-hoc calibration of pre-trained artificial intelligence (AI) models has become increasingly important as a means

---
\*Equal contribution  [1]IoT Thrust, The Hong Kong University of Science and Technology (Guangzhou), Guangzhou, China [2]Department of ECE, The Hong Kong University of Science and Technology, HK SAR [3]Department of Engineering, King's College London, London, U.K.. Correspondence to: Hong Xing <hongxing@ust.hk>.

*Proceedings of the 42^nd International Conference on Machine Learning*, Vancouver, Canada. PMLR 267, 2025. Copyright 2025 by the author(s).

to ensure reliable inference and decision-making in safety-critical domains such as healthcare (Kompa et al., 2021), engineering (Cohen et al., 2023) and large language models (LLMs) (Ji et al., 2023; Huang et al., 2025). Conformal prediction (CP) is a model-agnostic post-hoc calibration framework that provides distribution-free statistical coverage guarantees. This is done by augmenting an AI model's decisions with prediction sets evaluated on the basis of held-out data (Vovk et al., 2005; Angelopoulos et al., 2024b).

CP uses held-out calibration data to infer statistical information about the distribution of the errors made by the pre-trained AI model. Based on this analysis, CP associates to each decision of the AI model a prediction set that includes all the outputs that are consistent with the inferred error statistics. Specifically, CP calculates a quantile of performance scores attained by the pre-trained model on the calibration data (Vovk et al., 2005; Lei et al., 2018; Barber et al., 2021). The prediction set provably meets marginal coverage guarantees for a user-defined target. Accordingly, the correct output is included in the prediction set with high probability with respect to the distribution of calibration and test data.

In light of its simplicity and of its strong theoretical properties, CP has been recently developed in several directions, including to address more general risk functions (Angelopoulos et al., 2024c), to provide localized statistical guarantees (Gibbs et al., 2025), to operate via an online feedback-based mechanism (Gibbs & Candes, 2021), and to incorporate Bayesian inference mechanisms (Clarté & Zdeborová, 2024). Furthermore, it has been applied in safety-critical areas such as medical diagnostics (Lu et al., 2022) and LLMs (Quach et al., 2024; Mohri & Hashimoto, 2024).

As mentioned, CP requires access to a data set of calibration data points. However, in practice, a decision maker may not have sufficient calibration data stored locally. This is an important issue, since the size of the prediction set depends on the number of available calibration data points, and thus a small calibration data set would yield uninformative prediction sets (Zecchin et al., 2024). However, calibration data may be available in a decentralized fashion across multiple devices (Xu et al., 2023). Examples include diagnostic healthcare models at different hospitals,

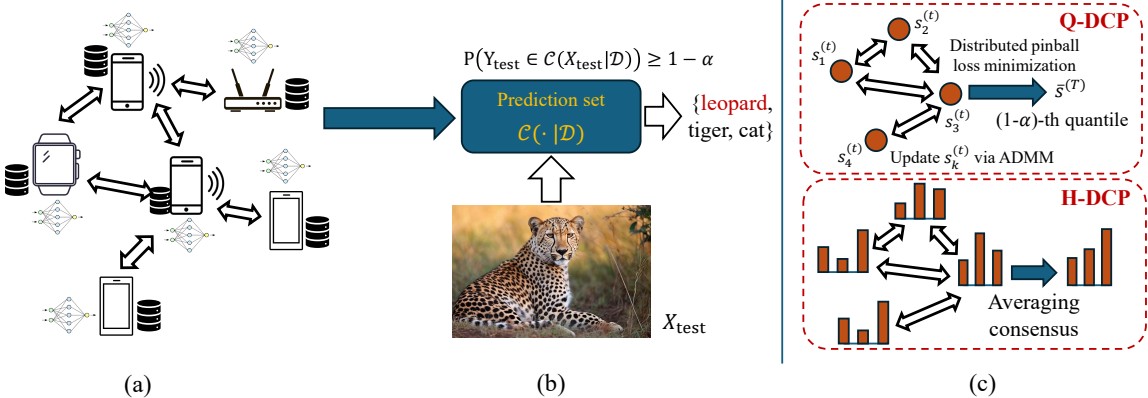

*Figure 1.* (a) This work studies a decentralized inference setting in which multiple devices share the same pre-trained model, and each device has a local calibration data set. (b) Given a common input, the devices aim at producing a prediction set that includes the true label with a user-defined probability $1 - \alpha$. (c) We propose two message-passing schemes for conformal prediction (CP): Quantile-based distributed CP (Q-DCP) addresses the decentralized optimization of the pinball, or quantile, loss over the calibration scores; while histogram-based distributed CP (H-DCP) targets the consensus-based estimate of the histogram of the calibration scores.

Internet-of-Things (IoT) systems, and autonomous vehicle networks. In many of these scenarios, the distributed devices are privacy-conscious, preventing a direct exchange of the local calibration data sets.

With this motivation, prior art has studied settings in which multiple data-holding devices are connected to the decision-making device via capacity-limited links in a star topology (Humbert et al., 2023; Lu et al., 2023; Plassier et al., 2023; Zhu et al., 2024), which is widely adopted for federated learning (McMahan et al., 2017; Kairouz et al., 2021). In these systems, the central device collects information from the devices to estimate the relevant global quantile of the calibration data distributed across the devices. In particular, in (Humbert et al., 2023), the global quantile is estimated via a quantile-of-quantiles approach, in which the devices transmit their local quantiles to the central server and the quantile of the received quantiles is calculated at the central server side. In (Zhu et al., 2024), the central server estimates the average of the local histograms of quantized calibration scores and then estimates the global quantile based on the average histogram.

### 1.2. Main Contributions

The star topology assumed in the prior art does not reflect many important scenarios of interest in which communication is inherently local, being limited to the neighbors of each device. As illustrated in Fig. 1(a), this paper studies CP in a fully decentralized architecture in which the data-holding devices can only communicate via message passing on a connectivity graph. In this setting, all devices are decision-makers that have access to limited local calibration data and share a common pre-trained model. As shown in Fig. 1(b), given a common input, the devices aim

at producing a prediction set that includes the true label with a user-defined probability.

This paper proposes two distributed CP (DCP) approaches: quantile-based DCP (Q-DCP) and histogram-based DCP (H-DCP). Q-DCP employs distributed quantile regression enhanced with tailored smoothing and regularization terms to accelerate convergence via message passing, while H-DCP uses a consensus-based histogram estimation approach inspired by (Zhu et al., 2024).

Specifically, the main contributions of this work are as follows:

1. We introduce Q-DCP, a message-passing CP protocol based on quantile regression. Q-DCP addresses a decentralized quantile regression problem by means of the alternating direction method of multipliers (ADMM) (Boyd et al., 2011) with tailored smoothing and regularization to accelerate the convergence speed. Q-DCP guarantees marginal coverage as centralized CP, as long as hyperparameters related to the network topology and to the initialization are properly selected.

2. We also introduce H-DCP, a decentralized CP protocol that attains hyperparameter-free coverage guarantees. H-DCP estimates the global histogram of quantized local scores via a consensus algorithm.

3. Experiments explore the trade-offs between hyperparameter tuning requirements, communication overhead, coverage guarantees, and prediction set sizes for Q-DCP and H-DCP across different network topologies. While H-DCP requires a larger communication load than Q-DCP, its coverage guarantees are not tied to hyperparameter selection.

## 1.3. Notations

We use lower-case letter $x$ to denote scalars; upper-case letter $X$ to denote random variables; bold lower-case letter $\boldsymbol{x}$ to denote vectors; bold upper-case letter $\boldsymbol{X}$ to denote matrices; script letter $\mathcal{X}$ to denote sets. We denote $\|\boldsymbol{x}\|$ as the $l_2$-norm of a vector $\boldsymbol{x}$; $\mathbb{I}\{\cdot\}$ as the indicator function; $\|\boldsymbol{x}\|_G = \sqrt{\boldsymbol{x}^T \boldsymbol{G} \boldsymbol{x}}$ as the $G$-norm of a vector $\boldsymbol{x}$ with a positive semidefinite matrix $\boldsymbol{G}$; $\sigma_{\max}(\boldsymbol{X})$ and $\sigma_{\min}(\boldsymbol{X})$ as the largest singular value and the smallest non-zero singular value of a matrix $\boldsymbol{X}$, respectively; and $\lambda_i(\boldsymbol{X})$ to denote the $i$-th largest eigenvalue of a matrix $\boldsymbol{X}$.

## 2. Problem Description

We study a network of $K$ devices in which each device $k$ has access to a local calibration data set $\mathcal{D}_k = \{(X_{i,k}, Y_{i,k})\}_{i=1}^{n_k}$ with data points $(X_{i,k}, Y_{i,k})$ drawn i.i.d. from a distribution $P_k$. We define the global data set as the union $\bigcup_{k=1}^{K} \mathcal{D}_k = \mathcal{D}$ and $n = \sum_{k=1}^{K} n_k$ as the total number of data points. Using the local data sets $\{\mathcal{D}_k\}_{k=1}^{K}$, and message-passing-based communication, the devices aim to collaboratively calibrate the decision of a shared pre-trained model $f : \mathcal{X} \mapsto \mathcal{Y}$.

To elaborate, assume that a test input $X_{\text{test}}$ is available at all devices. This may represent, e.g., a common observation in a sensor network or a user query distributed to multiple servers. Given a target coverage level $(1 - \alpha)$ for $\alpha \in [0, 1]$, the goal of the system is to determine a set-valued function $\mathcal{C}(\cdot|\mathcal{D}) : \mathcal{X} \mapsto 2^{\mathcal{Y}}$ with marginal coverage guarantees. Specifically, following (Lu et al., 2023), given a test data point $(X_{\text{test}}, Y_{\text{test}})$, drawn from the mixture distribution

$$P = \sum_{k=1}^{K} w_k P_k \tag{1}$$

for some weight $w_k \geq 0$ and $\sum_{k=1}^{K} w_k = 1$, we impose the coverage requirement

$$P(Y_{\text{test}} \in \mathcal{C}(X_{\text{test}}|\mathcal{D})) \geq 1 - \alpha. \tag{2}$$

This condition requires that the test label $Y_{\text{test}}$ belongs to the prediction set $\mathcal{C}(X_{\text{test}}|\mathcal{D})$ with probability at least $1 - \alpha$. The probability in (2) is evaluated over the calibration data in $\mathcal{D}$ and the test data $(X_{\text{test}}, Y_{\text{test}})$. As in (Lu et al., 2023), we will specifically concentrate on the choice

$$w_k \propto n_k + 1, \tag{3}$$

in which the weight $w_k$ for each device $k$ is proportional to the size of the local data set $\mathcal{D}_k$.

The average size of the output of $\mathcal{C}(\cdot|\mathcal{D})$, referred to as the *inefficiency*, is defined as the expectation $\mathbb{E}|\mathcal{C}(X_{\text{test}}|\mathcal{D})|$, where the average is evaluated over $X_{\text{test}}$ and on the distribution of the data $\mathcal{D}$. Ideally, the prediction set $\mathcal{C}(X_{\text{test}}|\mathcal{D})$

minimizes the inefficiency while satisfying the constraint (2).

As shown in Fig. 1(a), in order to produce the prediction set $\mathcal{C}(X_{\text{test}}|\mathcal{D})$, devices can communicate over a connectivity graph. The connectivity graph $\mathcal{G}(\mathcal{V}, \mathcal{E})$ is undirected, with $\mathcal{V}$ denoting the set of devices with $|\mathcal{V}| = K$ and $\mathcal{E} \subseteq \{(i, j) \in \mathcal{V} \times \mathcal{V} | i \neq j\}$ denoting the set of edges with $|\mathcal{E}| = E$. The connectivity of the graph is characterized by the $2E \times K$ incidence matrix $\boldsymbol{A} = [\boldsymbol{A}_1^T, \boldsymbol{A}_2^T]^T$ with submatrices $\boldsymbol{A}_1, \boldsymbol{A}_2 \in \mathbb{R}^{E \times K}$. Denoting the index corresponding to an edge $(i, j) \in \mathcal{E}$ by $q \in \{1, \ldots, |\mathcal{E}|\}$, the $(q, i)$-th entry of matrix $\boldsymbol{A}_1$ and the $(q, j)$-th entry of matrix $\boldsymbol{A}_2$ equal 1 if $(i, j)$ is an edge in graph $\mathcal{G}(\mathcal{V}, \mathcal{E})$, i.e., $(i, j) \in \mathcal{E}$, and they equal zero otherwise. Each device $k$ can only communicate with its set of neighbors, denoted as $\mathcal{N}_k = \{j \in \mathcal{V} | (j, k) \in \mathcal{E}\}$, under communication constraints to be specified in Sec. 4.

## 3. Background

### 3.1. Split Conformal Prediction

Split CP provides a general framework for the design of post-hoc calibration strategies satisfying the coverage requirement (2) in centralized settings. CP constructs the prediction set $\mathcal{C}(\cdot|\mathcal{D})$ based on a calibration data set $\mathcal{D}$. This is done by evaluating a quantile, dependent on the largest miscoverage level $\alpha$, of the scores assigned by the pre-trained model $f$ to the calibration data points in set $\mathcal{D}$.

To elaborate, define as $s(x, y)$ a negatively oriented score derived from model $f$, such as the absolute error $s(x, y) = |y - f(x)|$ for regression or the log-loss $s(x, y) = -\log f_y(x)$ for classification. Write as $S_i = s(X_i, Y_i)$ the score assigned by the model $f$ to the $i$-th calibration data point $(X_i, Y_i)$ in the data set $\mathcal{D}$. The prediction set is evaluated by including all the labels $y \in \mathcal{Y}$ with a score no larger than a fraction, approximately $1 - \alpha$, of the calibration scores $\{S_i\}_{i=1}^{n}$. Specifically, CP produces the set

$$\mathcal{C}(X|\mathcal{D}) = \{y \in \mathcal{Y} : s(X, y) \leq \\ Q((1 - \alpha)(1 + 1/n); \{S_i\}_{i=1}^{n})\}, \tag{4}$$

where $Q(\gamma; \{S_i\}_{i=1}^{n})$ is the $\lceil \gamma \rceil$-th smallest value of the set $\{S_i\}_{i=1}^{n}$, which is set as the score threshold.

The empirical $\gamma$-quantile $Q(\gamma; \{S_i\}_{i=1}^{n})$ in (4) can be obtained by solving the following quantile regression problem (Koenker & Bassett Jr, 1978)

$$Q(\gamma; \{S_i\}_{i=1}^{n}) = \arg\min_{s \in \mathbb{R}} \rho_\gamma(s \,|\{S_i\}_{i=1}^{n}), \tag{5}$$

where $\rho_\gamma(\cdot \,|\{S_i\}_{i=1}^n)$ is the pinball loss function defined as

$$
\begin{aligned}
\rho_\gamma(s \,|\{S_i\}_{i=1}^n) = \\
\gamma \sum_{i=1}^n \mathrm{ReLU}(S_i - s) + (1-\gamma) \sum_{i=1}^n \mathrm{ReLU}(s - S_i),
\end{aligned} \tag{6}
$$

with $\mathrm{ReLU}(x) = \max(x, 0)$.

### 3.2. Distributed Conformal Prediction on a Star Topology

The calculation of the empirical quantile in the prediction set (4) requires full knowledge of the scores of all calibration data points in set $\mathcal{D}$. When the data points are distributed across privacy-sensitive and communication-constrained devices as in the setting under study in this paper, evaluating the empirical quantile is not possible unless communication is enabled. Prior work has studied a federated setting in which all devices are connected to a centralized server (Humbert et al., 2023; Lu et al., 2023; Plassier et al., 2023; Zhu et al., 2024; 2025; Kang et al., 2024). For reference, we briefly introduce FedCP-QQ (Humbert et al., 2023), FCP (Lu et al., 2023) and WFCP (Zhu et al., 2024) next.

**FedCP-QQ:** Denote as $\{S_{i,k} = s(X_{i,k}, Y_{i,k})\}_{i=1}^{n_k}$ the scores evaluated using the shared model $f$ on the local data set $\mathcal{D}_k$ for device $k$. In FedCP-QQ (Humbert et al., 2023), each device $k$ first calculates the empirical $(1 - \alpha')$-quantile of its local scores, which is transmitted to the centralized server for some probability $\alpha'$. After collecting $K$ local quantiles, the central server calculates the empirical $(1 - \beta)$-quantile of the received quantiles, thus obtaining quantile-of-quantiles. By optimizing the probabilities $\alpha'$ and $\beta$, the prediction set (4) is constructed using quantile-of-quantiles as the threshold. This approach satisfies the coverage condition (2) if the data sets $\mathcal{D}_k$ are i.i.d. across devices.

**FCP:** While FedCP-QQ assumes i.i.d. data sets across devices, FCP (Lu et al., 2023) adopts the model described in Sec. 2 allowing data points from different local data sets to be drawn from different distributions $\{P_k\}_{k=1}^K$, while the test data point $(X_{\text{test}}, Y_{\text{test}})$ is drawn from a mixture $P = \sum_{k=1}^K w_k P_k$ of the local distributions. In FCP, the central server collects the local scores from the $K$ devices, and then calculates the empirical $(1-\alpha)(1+K/n)$-quantile. This value is used as the threshold to construct prediction set $\mathcal{C}(X|\mathcal{D})$ in (4). FCP satisfies coverage condition (2) by choosing the weight $w_k$ to be proportional to $n_k + 1$, i.e., $w_k \propto n_k + 1$.

**WFCP:** Unlike FedCP-QQ and FCP, which communicate quantiles between devices and server, Zhu et al. (2024) proposed to exchange information about the local histograms of the quantized scores. Specifically, the scores $\{S_{i,k}\}_{i=1}^{n_k}$ are first quantized to $M$ levels by each device $k$, which then evaluates the histogram vector $\boldsymbol{p}_k = [p_{1,k}, p_{2,k}, \ldots, p_{M,k}]^T \in \mathbb{R}^M$ of the scores. The vectors $\boldsymbol{p}_k$, with $k = 1, 2, ..., K$, are synchronously transmitted to the centralized server on an additive multi-access channel, and the server estimates the average histogram $\bar{\boldsymbol{p}} = \frac{1}{K} \sum_{k=1}^K \boldsymbol{p}_k$ based on the received signal. The empirical $(1 - \alpha')$-quantile of all the scores is then estimated from the estimated average histogram $\bar{\boldsymbol{p}}$ for some optimized value $\alpha'$. Under the non-i.i.d. setting described in Sec. 2, WFCP can guarantee the coverage condition (2).

## 4. Quantile-based Distributed CP

In this section, we present the first fully decentralized protocol proposed in this work, which is referred to as quantile-based distributed CP (Q-DCP).

### 4.1. Decentralized Quantile Regression via ADMM

Q-DCP addresses the quantile regression problem (5) in the fully decentralized setting described in Sec. 2 using ADMM (Boyd et al., 2011). This strategy obtains an approximation of the empirical quantile $\mathrm{Q}((1 - \alpha)(1 + K/n); \{S_i\}_{i=1}^n)$ with a controlled error, which requires a single scalar broadcast to the neighbors of each device at each optimization iteration. As will be discussed, on the flip side, Q-DCP requires careful hyperparameter tuning in order to satisfy the coverage condition (2). The alternative strategy proposed in the next section alleviates such requirements on hyperparameter tuning at the cost of a larger per-iteration communication cost.

Formally, in Q-DCP, the $K$ devices collaboratively solve the quantile regression problem

$$
\underset{s \in \mathbb{R}}{\text{Minimize}} \quad \sum_{k=1}^K \rho_{(1-\alpha)(1+\frac{K}{n})}\left(s \,|\{S_{i,k}\}_{i=1}^{n_k}\right), \tag{7}
$$

where target coverage $(1 - \alpha)(1 + K/n)$ follows FCP (Lu et al., 2023) (see previous section). The objective function (7) is convex, but it is not smooth or strongly convex. For such an objective function, a direct application of ADMM would exhibit a sub-linear convergence rate, causing a large optimality gap when the number of communication rounds is limited (He & Yuan, 2012).

To ensure a linear convergence rate, we propose to replace the ReLU operation in (6) with the smooth function $\tilde{g}(x) = x + (1/\kappa) \log(1 + e^{-\kappa x})$, where smaller $\kappa$ leads to greater smoothness at the cost of larger approximation error. In practice, the function $\tilde{g}(x)$ coincides with $\mathrm{ReLU}(x)$ as $\lim_{\kappa \to \infty} \tilde{g}(x) = \mathrm{ReLU}(x)$ (Chen & Mangasarian, 1996). Furthermore, to guarantee strong convexity, we add a regu-

larization term to the pinball loss function (6) as

$$\tilde{\rho}_\gamma(s \mid \{S_i\}_{i=1}^n) = \gamma \sum_{i=1}^n \tilde{g}(S_i - s)$$
$$+ (1 - \gamma) \sum_{i=1}^n \tilde{g}(s - S_i) + \frac{\mu}{2}(s - s_0)^2, \quad (8)$$

where $\mu > 0$ and $s_0$ are hyperparameters. The modified pinball loss function $\tilde{\rho}_\gamma(\cdot \mid \{S_i\}_{i=1}^n)$ is *L-smooth and $\mu$-strongly convex*, with $L = (n\kappa)/4 + \mu$. Using the smooth and strongly convex loss in (8), the decentralized optimization of quantile regression in (7) is modified as

$$\underset{s \in \mathbb{R}}{\text{Minimize}} \quad \sum_{k=1}^K \tilde{\rho}_{(1-\alpha)(1+\frac{K}{n})} \left(s \mid \{S_{i,k}\}_{i=1}^{n_k}\right). \quad (9)$$

Note that the (unique) solution of problem (9), denoted by $\hat{s}^*$, is generally different from the optimal solution $s^*$ of problem (7), unless $\kappa$ tends to infinity and $\mu$ is set to zero.

Q-DCP adopts ADMM to address problem (9), obtaining the formulation

$$\underset{s \in \mathbb{R}^K, z \in \mathbb{R}^E}{\text{Minimize}} \quad \tilde{f}(s) = \sum_{k=1}^K \tilde{\rho}_{(1-\alpha)(1+\frac{K}{n})} \left(s_k \mid \{S_{i,k}\}_{i=1}^{n_k}\right)$$

$$\text{Subject to} \quad As + Bz = 0,$$
$$(10)$$

where $A \in \mathbb{R}^{2E \times K}$ is the incidence matrix defined in Sec. 2, $B = [-I_E; -I_E] \in \mathbb{R}^{2E \times E}$, and $z$ is an auxiliary variable imposing the consensus constraint on neighboring devices, i.e., $s_i = z_q$ and $s_j = z_q$ if $(i, j) \in \mathcal{E}$ with $q$ being the corresponding index of edge $(i, j)$.

### 4.2. Description of Q-DCP

Following ADMM (Boyd et al., 2011), Q-DCP solves problem (10) by considering the augmented Lagrangian

$$L_c(s, z, \lambda) = \tilde{f}(s) + \langle \lambda, As + Bz \rangle + \frac{c}{2}\|As + Bz\|_2^2, \quad (11)$$

where $\lambda \in \mathbb{R}^{2E}$ is the Lagrange multiplier associated with the $2E$ equality constraints in (10), and $c > 0$ is a hyperparameter. The updates of the local estimated quantiles $s$, the consensus variables $z$ and the dual variables $\lambda$ at iteration $t + 1$ are given by (Boyd et al., 2011)

$$s^{(t+1)} = \underset{s \in \mathbb{R}^K}{\operatorname{argmin}} \, L_c(s, z^{(t)}, \lambda^{(t)}), \quad (12a)$$

$$z^{(t+1)} = \underset{z \in \mathbb{R}^E}{\operatorname{argmin}} \, L_c(s^{(t+1)}, z, \lambda^{(t)}), \quad (12b)$$

$$\lambda^{(t+1)} = \lambda^{(t)} + c\left(As^{(t+1)} + Bz^{(t+1)}\right). \quad (12c)$$

After $T$ iterations, ADMM produces the quantile estimate $s_k^{(T)}$ for each device $k \in \mathcal{V}$. Then, the devices run the fast

distributed averaging algorithm (Xiao & Boyd, 2004) to obtain the average $\bar{s}^{(T)} = 1/K \sum_{k \in \mathcal{V}} s_k^{(T)}$ with negligible error.

Finally, the prediction set is constructed as

$$\mathcal{C}^{\text{Q-DCP}}(X|\mathcal{D}) = \left\{y \in \mathcal{Y} : s(X, y) \leq s^{\text{Q-DCP}}\right\}, \quad (13)$$

with

$$s^{\text{Q-DCP}} = \bar{s}^{(T)} + \epsilon^{\text{Q-DCP}}, \quad (14)$$

where $\epsilon^{\text{Q-DCP}}$ is an upper bound on the error $|\bar{s}^{(T)} - s^*|$ of the quantile estimate $\bar{s}^{(T)}$ to be derived in the next subsection.

The proposed Q-DCP is summarized in Algorithm 1.

---

**Algorithm 1** Q-DCP

**Input:** ADMM parameters $c > 0$ and $b > 1$, number of iterations $T$, incidence matrix $A$, smoothness parameter $L$ and regularization parameters $(\mu, s_0, \epsilon_0)$, coverage level $1 - \alpha$, and score $s(\cdot, \cdot)$
Initialize $s^{(0)} = z^{(0)} = \lambda^{(0)} = 0$ and $t = 0$
▷ ADMM (Boyd et al., 2011):
**while** $t < T$ **do**
  Update $s^{(t+1)}$ by solving

$$\nabla f(s^{(t+1)}) + A^T \lambda^{(t)} + cA^T(As^{(t+1)} + Bz^{(t)}) = 0$$

  $z^{(t+1)} = -(cB^T B)^{-1} B^T (cAs^{(t+1)} + \lambda^{(t)})$
  $\lambda^{(t+1)} = \lambda^{(t)} + cAs^{(t+1)} + cBz^{(t+1)}$
  $t \leftarrow t + 1$
**end while**
▷ Prediction set construction:
**for** device $k \in \mathcal{V}$ **do**
  Run distributed averaging (Xiao & Boyd, 2004) to obtain $\bar{s}^{(T)} = \frac{1}{K} \sum_{k=1}^K s_k^{(T)}$
  Calculate $\epsilon^{(T)}$ and $\tilde{\epsilon}_0$ using (17) and (16), respectively, and set $s^{\text{Q-DCP}} = \bar{s}^{(T)} + \epsilon^{(T)} + \tilde{\epsilon}_0$
**end for**
**Output:** $\mathcal{C}^{\text{Q-DCP}}(\cdot|\mathcal{D}) = \left\{y \in \mathcal{Y} : s(\cdot, y) \leq s^{\text{Q-DCP}}\right\}$

---

### 4.3. Theoretical Guarantees

In this section, we first derive the $\epsilon^{\text{Q-DCP}}$ upper bound on the estimation error of Q-DCP used in constructing the prediction set (13). Then we prove that Q-DCP attains the coverage guarantee (2).

By the triangle inequality, the estimation error $|\bar{s}^{(T)} - s^*|$ can be upper bounded as

$$|\bar{s}^{(T)} - s^*| \leq |\bar{s}^{(T)} - \hat{s}^*| + |\hat{s}^* - s^*|, \quad (15)$$

where the first term accounts for the convergence error for problem (10), while the second term quantifies the bias caused by the use of the smooth and strongly convex approximation (8).

The bias term can be bounded as follows. We start with the following assumption.

**Assumption 4.1.** The regularization parameter $s_0$ and the optimal solution $s^*$ in (5) differ by at most $\epsilon_0 \geq 0$, i.e., $|s_0 - s^*| \leq \epsilon_0$.

**Proposition 4.2.** *Under Assumption 4.1, the bias $|\hat{s} - s^*|$ is upper bounded as*

$$|s^* - \hat{s}^*| \leq \sqrt{\frac{2n \log(2)}{\mu \kappa} + \epsilon_0^2} = \tilde{\epsilon}_0. \qquad (16)$$

*Proof:* See supplementary material (see Appendix A.1). □

Next, the convergence error $|\bar{s}^{(T)} - \hat{s}^*|$ is bounded by leveraging on existing results on the convergence of ADMM.

**Proposition 4.3** (Theorem 1, Shi et al., 2014)**.** *The convergence error $|\bar{s}^{(T)} - \hat{s}^*|$ is upper bounded as*

$$|\bar{s}^{(T)} - \hat{s}^*| \leq \sqrt{\frac{1}{K\mu} \left( \frac{1}{1+\delta} \right)^{T-1}} \|\boldsymbol{u}^{(0)} - \boldsymbol{u}^*\|_G \qquad (17)$$
$$= \epsilon^{(T)},$$

*where the parameter $\delta > 0$ is defined as*

$$\delta = \min \big\{ \left( (b-1)\sigma_{\min}^2(\boldsymbol{M}_-) \right) / \left( b\sigma_{\max}^2(\boldsymbol{M}_+) \right), \\ \mu / \left( (c/4)\sigma_{\max}^2(\boldsymbol{M}_+) + (b/c)L^2\sigma_{\min}^{-2}(\boldsymbol{M}_-) \right) \big\}, \qquad (18)$$

*with any $b > 1$, $\boldsymbol{M}_- = \boldsymbol{A}_1^T - \boldsymbol{A}_2^T$, $\boldsymbol{M}_+ = \boldsymbol{A}_1^T + \boldsymbol{A}_2^T$, and $\boldsymbol{G} = \begin{pmatrix} c\boldsymbol{I}_E & 0_E \\ 0_E & (1/c)\boldsymbol{I}_E \end{pmatrix}$. Moreover, $\boldsymbol{u}^{(0)} = [(\boldsymbol{z}^{(0)})^T, (\tilde{\boldsymbol{\lambda}}^{(0)})^T]^T$ with $\tilde{\boldsymbol{\lambda}}^{(0)} \in \mathbb{R}^E$ containing the first $E$ entries of the vector $\boldsymbol{\lambda}^{(0)}$, and we also write as $\boldsymbol{u}^* = [(\boldsymbol{z}^*)^T, (\tilde{\boldsymbol{\lambda}}^*)^T]^T$ the $2E \times 1$ vector collecting of the optimal primal solution $\boldsymbol{z}^*$ and the optimal Lagrange multipliers $\tilde{\boldsymbol{\lambda}}^*$ of problem (10).*

The upper bound (17) depends on the initial error, quantified by the distance $\|\boldsymbol{u}^{(0)} - \boldsymbol{u}^*\|_G$, and by the connectivity of the graph. In fact, a larger connectivity is reflected by a larger ratio $\sigma_{\min}(\boldsymbol{M}_-)/\sigma_{\max}(\boldsymbol{M}_+)$ (Fiedler, 1973), and thus a larger parameter $\delta$ in (17).

Using the obtained bounds (16) and (17) in (15), we obtain the following main result on the performance for Q-DCP.

**Theorem 4.4** (Coverage Guarantee for Q-DCP)**.** *The prediction set $\mathcal{C}^{Q\text{-}DCP}(\cdot|\mathcal{D})$ in (13) produced by Q-DCP satisfies the coverage condition (2) with weights selected as in (3).*

*Proof:* See supplementary material (see Appendix A.2) □

## 5. Histogram-based Distributed CP

As demonstrated by Theorem 4.4, Q-DCP requires knowledge of a parameter $\epsilon_0$ satisfying Assumption 4.1, as well

as of the parameter $\delta$ in (17), which depends on the connectivity properties of the graph through the incidence matrix $\boldsymbol{A}$. In this section, we present an alternative fully decentralized protocol, referred to as histogram-based DCP (H-DCP), which provides rigorous coverage guarantees without the need for hyperparameter tuning as in Q-DCP, but at the cost of larger communication overhead.

In H-DCP, devices exchange histograms of quantized calibration scores in a manner similar to WFCP (Zhu et al., 2024). As a result of a consensus algorithm, the devices evaluate the average histogram in order to yield an estimate of the mixture distribution P in (1) with weights (3). From this estimate, a suitable bound is derived on the $(1 - \alpha)(1 + K/n)$-quantile of the distribution P to evaluate the prediction set of the form (4).

### 5.1. Description of H-DCP

To start, in H-DCP, all devices apply a uniform scalar quantizer, $\Gamma(\cdot) : [0,1] \mapsto \{1/M, 2/M, \ldots, 1\}$, with step size $1/M$, to all the local calibration scores under the following assumption.

**Assumption 5.1.** The score function $s(\cdot, \cdot)$ is bounded, i.e., without loss of generality, $0 \leq s(\cdot, \cdot) \leq 1$.

The quantizer is formally defined as

$$\Gamma(s) = \begin{cases} \frac{1}{M} & s \in [0, \frac{1}{M}] \\ \frac{m}{M} & s \in \left( \frac{m-1}{M}, \frac{m}{M} \right] \end{cases}, \text{ for } m = 2, \ldots, M. \qquad (19)$$

With this quantizer, each device $k \in \mathcal{V}$ evaluates the local histogram $\boldsymbol{p}_k = [p_{1,k}, \ldots, p_{M,k}]^T$ associated with the local quantized scores $\{\Gamma(S_{i,k})\}_{i=1}^{n_k}$, with $\sum_{m=1}^M p_{m,k} = 1$. Accordingly, this vector includes the entries

$$p_{m,k} = \frac{1}{n_k} \sum_{i=1}^{n_k} \mathbb{I} \left\{ \Gamma(S_{i,k}) = \frac{m}{M} \right\} \qquad (20)$$

with $p_{m,k}$ representing the fraction of the quantized scores associated with the $m$-th quantization level at device $k \in \mathcal{V}$. The sum

$$p_m = \frac{1}{n} \sum_{k=1}^K n_k p_{m,k} = \frac{1}{n} \sum_{k=1}^K \sum_{i=1}^{n_k} \mathbb{I} \left\{ \Gamma(S_{i,k}) = \frac{m}{M} \right\} \qquad (21)$$

corresponds to the fraction of quantized scores equal to $m/M$ in the set of quantized scores for the global data set $\mathcal{D}$, defining the global histogram vector $\boldsymbol{p} = [p_1, \ldots, p_M]$.

In H-DCP, the devices aim at estimating the global score histogram $\boldsymbol{p}$ in (21) in order to obtain an estimate of the $(1 - \alpha)(1 + K/n)$-quantile of the quantized calibration

scores, i.e.,

$$\frac{m_\alpha}{M} = Q\left((1-\alpha)(1+K/n); \cup_{k=1}^K \{\Gamma(S_{i,k})\}_{i=1}^{n_k}\right)$$

$$= \frac{1}{M} \argmin_{m=1,\dots,M} \left\{ \sum_{i=1}^m p_i \geq (1-\alpha)\left(1+\frac{K}{n}\right) \right\}. \quad (22)$$

By the design of the quantizer (19), this quantile provides an upper bound on the $(1-\alpha)(1+K/n)$-quantile of the unquantized calibration score.

To estimate the sum (21), the devices apply consensus with a matrix $\boldsymbol{W} \in \mathbb{R}^{K \times K}$ satisfying $\boldsymbol{W} = \boldsymbol{W}^T$, $\boldsymbol{W}\boldsymbol{1} = \boldsymbol{1}^T\boldsymbol{W} = \boldsymbol{1}^T$ and $\|\boldsymbol{W} - \boldsymbol{1}\boldsymbol{1}^T/K\|_2 < 1$ with entries $[\boldsymbol{W}]_{k,j} = W_{k,j} > 0$ if $(k,j) \in \mathcal{E}$ and $[\boldsymbol{W}]_{k,j} = 0$ otherwise. Following the fast distributed averaging algorithm (Xiao & Boyd, 2004), each device $k \in \mathcal{V}$ updates the local vector $\boldsymbol{x}_k$ by a linear consensus step

$$\boldsymbol{x}_k^{(t+1)} = \boldsymbol{x}_k^{(t)} + \eta \sum_{j=1}^K W_{kj}(\boldsymbol{x}_j^{(t)} - \boldsymbol{x}_k^{(t)}), \quad (23)$$

where we define the initialization $\boldsymbol{x}_k^{(0)} = \boldsymbol{x}_k = ((Kn_k)/n)\boldsymbol{p}_k$ such that $\boldsymbol{p} = 1/K \sum_{k=1}^K \boldsymbol{x}_k$, and $\eta \in (0,1]$ is the update rate.

Finally, H-DCP constructs the prediction set at device $k$ as

$$C_k^{\text{H-DCP}}(X|\mathcal{D}) = \left\{ y \in \mathcal{Y} : \Gamma(s(X,y)) \leq \frac{m_{\alpha,k}^{\text{H-DCP}}}{M} \right\}, \quad (24)$$

where

$$m_{\alpha,k}^{\text{H-DCP}} = \argmin_{m=1,\dots,M} \left\{ \sum_{i=1}^m x_{i,k}^{(T)} \right. $$
$$\left. \geq (1-\alpha)\left(1+\frac{K}{n}\right) + \epsilon^{\text{H-DCP}} \right\}, \quad (25)$$

and $\epsilon^{\text{H-DCP}}$ is a parameter to be determined below to give an upper bound on the error $\|\boldsymbol{x}_k^{(T)} - \boldsymbol{p}\|$ for the local estimate $\boldsymbol{x}_k^{(T)}$ of the global vector $\boldsymbol{p}$ at device $k \in \mathcal{V}$ after $T$ global communication rounds. Note that if $(1-\alpha)(1+K/n) + \epsilon^{\text{H-DCP}} > 1$, we set $m_{\alpha,k}^{\text{H-DCP}} = M$.

The proposed H-DCP is summarized in Algorithm 2.

### 5.2. Theoretical Guarantee

In this subsection, we provide the main result on the coverage guarantee of H-DCP through the following theorem.

**Theorem 5.2** (Coverage Guarantee for H-DCP). *Setting*

$$\epsilon^{\text{H-DCP}} = \frac{K\sqrt{2KM}(1-\eta\varrho)^T}{n} \sqrt{\max_{k \in \mathcal{V}}\{n_k^2\} + \frac{1}{K}\sum_{j=1}^K n_j^2}, \quad (26)$$

---

**Algorithm 2** H-DCP

> **Input:** Consensus parameters $\eta > 0$ and matrix $\boldsymbol{W}$, number of iterations $T$, quantization level $M$, coverage level $1 - \alpha$, and score $s(\cdot, \cdot)$
> Each device $k \in \mathcal{V}$ calculates the local histogram $\boldsymbol{p}_k$ via (20)
> Initialize $\boldsymbol{x}_k^{(0)} = ((Kn_k)/n)\boldsymbol{p}_k$ and $t = 0$
> ▷ Consensus:
> **while** $t < T$ **do**
>   **for** device $k \in \mathcal{V}$ **do**
>     $\boldsymbol{x}_k^{(t+1)} = \boldsymbol{x}_k^{(t)} + \eta \sum_{j=1}^K W_{kj}(\boldsymbol{x}_j^{(t)} - \boldsymbol{x}_k^{(t)})$
>   **end for**
>   $t \leftarrow t + 1$
> **end while**
> ▷ Prediction set construction:
> **for** device $k \in \mathcal{V}$ **do**
>   Calculate $\epsilon^{\text{H-DCP}}$ via (26)
>   Calculate $m_{\alpha,k}^{\text{H-DCP}}$ via (25)
> **end for**
> **Output:** For all $k \in \mathcal{V}$, construct $C_k^{\text{H-DCP}}(\cdot|\mathcal{D})$ via (24)

---

*where $\varrho = 1 - |\lambda_2(\boldsymbol{W})|$ is the spectral gap of the consensus matrix $\boldsymbol{W}$, the prediction sets $\mathcal{C}_k^{\text{H-DCP}}(\cdot|\mathcal{D})$ in (24) produced by H-DCP for all $k \in \mathcal{V}$ satisfy the coverage condition (2) by choosing weights as in (3).*

*Proof:* See supplementary material (see Appendix A.3). □

Theorem 5.2 shows that H-DCP only requires knowledge of the spectral gap $\varrho$, which can be efficiently estimated in a fully decentralized manner (Muniraju et al., 2020). The flip side is that, while Q-DCP requires the exchange of a single real number at each iteration, H-DCP has a communication overhead $M$ times larger, requiring the exchange of an $M$-dimensional real vector.

## 6. Experimental Results

In this section, we report experimental results on the proposed decentralized CP protocols, Q-DCP and H-DCP, using the conventional centralized CP as a benchmark. We first study Q-DCP and H-DCP separately, and then provide a comparison between Q-DCP and H-DCP as a function of the communication load. Throughout, we evaluate the performance in terms of coverage and average set size, also known as inefficiency (Vovk et al., 2005), for different network topologies.

### 6.1. Setting

As in Lu et al. (2023), we first train a shared model $f(\cdot)$ using the Cifar100 training data set to generate the score function $s(\cdot, \cdot)$. Calibration data, obtained from the Cifar100 test data, is distributed in a non-i.i.d. manner among $K =$

20 devices by assigning 5 unique classes to each device. Specifically, device 1 receives $n_1$ data points corresponding to 0-4, device 2 receives $n_2$ data points from classes 5-9, and so on. We set $n_k = 50$ for all devices $k \in \mathcal{V}$. Using the shared model $f(\cdot)$, the score function is defined as $s(x, y) = 1 - [f(x)]_y$, where $[f(x)]_y$ is the confidence assigned by the model to label $y$ (Sadinle et al., 2019).

We extract the calibration data from the Cifar100 test data by sampling uniformly at random (without replacement), and average results are shown over 10 random generations of the calibration data. The set size is normalized to the overall number of classes, i.e., 100.

## 6.2. Results for Q-DCP

The hyperparameters for the Q-DCP loss (8) are chosen as follows. We set $\kappa = 2000$ for the smooth function $\tilde{g}(\cdot)$ as suggested by Nkansah et al. (2021), and we choose $\mu = 2000$. Moreover, unless noted otherwise, in (8), we set $s_0$ to be the average of the local score quantiles, which can be evaluated via a preliminary consensus step. Note that the need to choose a suitable value for $s_0$ is related to the general problem of hyperparameter-tuning for Q-DCP discussed in Sec. 4.3. This aspect will be further elaborated below by studying the impact of the selection of hyperparameter $\epsilon_0$ in (16).

The impact of the coverage level $1 - \alpha$ on Q-DCP is studied in Fig. 2 for $T = 1500$ and $\epsilon_0 = 0.1$, while results on convergence can be found in the supplementary material (see Appendix B.2.). We specifically consider the chain, cycle, star, torus, and complete graph, which are listed in order of increasing connectivity. Validating Theorem 4.4, the figure shows that Q-DCP provides coverage guarantees, with more conservative prediction sets obtained over graphs with lower connectivity. In particular, with a complete graph, the assumed $T = 1500$ iterations are seen to be sufficient to obtain the same set size as centralized CP.

As detailed in Theorem 4.4, Q-DCP requires a carefully selected hyperparameter $\epsilon_0$, satisfying Assumption 4.1, in order to meet coverage guarantees. To verify this point, Fig. 3 shows coverage and set size as a function of $1 - \alpha$ for the pair $(s_0 = -8 - 20\alpha, \epsilon_0 = 10^{-4})$, for which Assumption 4.1 is not satisfied. The figure confirms that Q-DCP can fail to yield coverage rates larger than $1 - \alpha$ for some topologies. This is particularly evident for graphs with stronger connectivity, which yield a faster rate of error compounding across the iterations.

## 6.3. Results for H-DCP

For H-DCP, unless noted otherwise, we set the consensus rate to $\eta = 1$, and the number of quantization levels to $M = 1000$. We choose the consensus matrix $\boldsymbol{W}$ in the

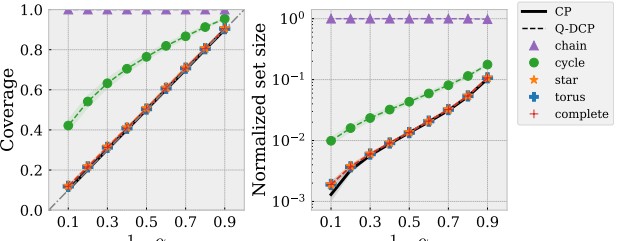

*Figure 2.* Coverage and normalized set size versus coverage level $1 - \alpha$ for CP and Q-DCP when Assumption 4.1 is satisfied ($T = 1500$ and $\epsilon_0 = 0.1$). The shaded error bars correspond to intervals covering 95% of the realized values.

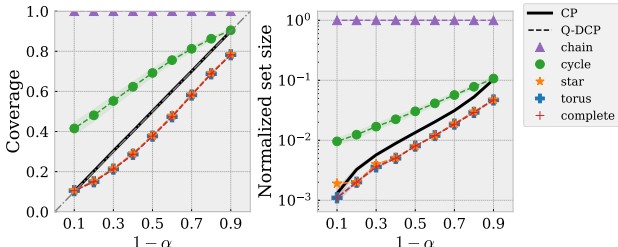

*Figure 3.* Coverage and normalized set size versus coverage level $1 - \alpha$ for CP and Q-DCP when Assumption 4.1 is not satisfied ($T = 1500$, $\epsilon_0 = 10^{-4}$, and $s_0 = -8 - 20\alpha$).

standard form with entries $W_{kj} = a$ for all edges $(k, j) \in \mathcal{E}$, $W_{kk} = 1 - |\mathcal{N}_k|a$ for all devices $k \in \mathcal{V}$, and $W_{kj} = 0$ otherwise, where $a = 2/(\lambda_1(\boldsymbol{L}) + \lambda_{K-1}(\boldsymbol{L}))$ with $\boldsymbol{L}$ being the Laplacian of the connectivity graph $\mathcal{G}$ (Xiao & Boyd, 2004). As illustrated in Fig. 4, unlike Q-DCP, H-DCP guarantees the coverage level $1 - \alpha$ without the need for hyperparameter tuning. Results on the convergence of H-DCP can be found in the supplementary material (see Appendix B.2).

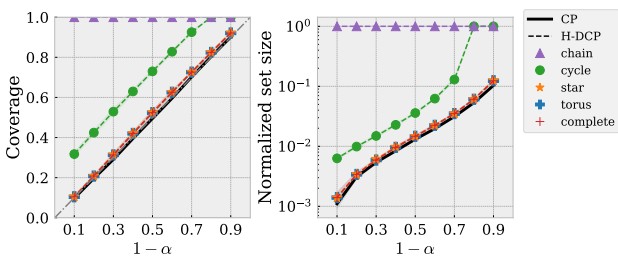

*Figure 4.* Coverage and normalized set size versus coverage level $1 - \alpha$ for CP and H-DCP ($T = 150$).

## 6.4. Comparing Q-DCP and H-DCP

Finally, we conduct experiments to compare the two proposed approaches in terms of the trade-offs among communication overhead, coverage guarantees, and prediction set sizes. In order to enable a fair comparison, we evaluate the performance of both Q-DCP and H-DCP under the same

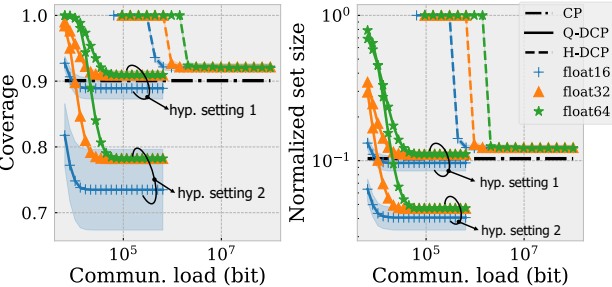

*Figure 5.* Coverage and normalized set size versus the total per-device communication load (torus graph with $\alpha = 0.1$, and Q-DCP with hyperparameter setting 1 given by ($\epsilon_0 = 0.1$, $s_0$ being the average of the local score quantile) and hyperparameter setting 2 given by ($\epsilon_0 = 10^{-4}$, $s_0 = -10$)).

total communication load per device. The communication load is evaluated in bits as $T \cdot C$, where $T$ is the number of iterations and $C$ denotes the number of bits communicated by one device per iteration. The per-iteration communication load $C$ is evaluated for the two schemes as detailed next.

For H-DCP, we fix the number of quantization levels to $M = 1000$, so that each histogram vector contains $M = 1000$ numbers. Each of these numbers is represented using a floating point format with $f \in \{16, 32, 64\}$ bits. Overall, the per-iteration per-device communication load of H-DCP is $C = Mf$. For Q-DCP, we also represent each estimated quantile $s_k^{(t)}$ using a floating point format with $f \in \{16, 32, 64\}$ bits, yielding a communication load equal to $C = f$.

Fig. 5 show the coverage and set size versus the total per-device communication load, $TC$, for both Q-DCP and H-DCP on the torus graph using different values of the numerical precision $f$. The choice of the torus graph is motivated by the fact that the spectral gap of the torus graph with 20 devices is moderate, providing an intermediate setting between a complete graph and a cycle graph. For Q-DCP, we consider two choices of hyperparameters, one, set as in Fig. 2, satisfying Assumption 4.1, and one, chosen as in Fig. 3, not satisfying this assumption.

As seen in Fig. 5, with well-selected hyperparameters, Q-DCP provides more efficient prediction sets, while also meeting the coverage requirement $1 - \alpha = 0.9$. An exception is given by the case $f = 16$, in which the reduced numerical precision of the inputs prevents Q-DCP from obtaining a high-quality solution for equation (12a). However, with poorly selected hyperparameters, Q-DCP can yield a violation of the coverage requirements even at a high precision $f$. In contrast, H-DCP maintains the coverage level $1 - \alpha$ across all considered communication loads.

# 7. Conclusion

This work has addressed the post-hoc calibration of pre-trained models in fully decentralized settings via conformal prediction (CP). We have proposed two message-passing-based approaches, quantile-based distributed CP (Q-DCP) and histogram-based distributed CP (H-DCP), that achieve reliable inference with marginal coverage guarantees. Q-DCP leverages distributed quantile regression with smoothing and regularization to enhance convergence, while H-DCP applies consensus-based histogram estimation. Our extensive experiments demonstrated the effectiveness of both methods in balancing communication overhead, coverage guarantees, and prediction set sizes across various network topologies. Specifically, Q-DCP was observed to have lower communication requirements, while being sensitive to hyperparameter tuning. In contrast, H-DCP offers robust coverage guarantees at the cost of a larger communication load.

Future work may investigate extension to asynchronous network (Wei & Ozdaglar, 2013; Tian et al., 2020), to localized risk guarantees (Gibbs et al., 2025; Zecchin & Simeone, 2024) and to online CP (Zhang et al., 2024; Angelopoulos et al., 2024a).

# Acknowledgements

The work of H. Xing was supported by the Guangdong Basic and Applied Basic Research Foundation under Grant 2025A1515010123, by the Guangdong Provincial Key Lab of Integrated Communication, Sensing and Computation for Ubiquitous Internet of Things under Grant 2023B1212010007, and by the Guangzhou Municipal Science and Technology Project under Grants 2024A04J4527 and 2024A03J0623. The work of O. Simeone was supported by the European Union's Horizon Europe project CENTRIC (101096379), by an Open Fellowship of the EPSRC (EP/W024101/1), and by the EPSRC project (EP/X011852/1).

# Impact Statement

This paper presents work whose goal is to advance the field of conformal prediction and reliable machine learning. There are many potential societal consequences of our work, none of which we feel must be specifically highlighted here.

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

# A. Proofs of Main Results

## A.1. Proof of Proposition 4.2

For simplicity, in this proof, we use the notations $\rho_\gamma(s) = \rho_\gamma(s|\{S_i\}_{i=1}^n)$ and $\tilde{\rho}_\gamma(s) = \tilde{\rho}_\gamma(s|\{S_i\}_{i=1}^n)$. By definition, we set $s^* = \arg\min_{s\in\mathbb{R}} \rho_\beta(s)$ and $\hat{s}^* = \arg\min_{s\in\mathbb{R}} \tilde{\rho}_\beta(s)$, where $\beta = (1-\alpha)(1+K/n)$. First, by $0 \leq \tilde{g}(x) - \max(x,0) \leq \frac{\log 2}{\kappa}$, we have

$$
\begin{aligned}
&\tilde{\rho}_\beta(s) - \rho_\beta(s) \\
&= \beta \sum_{i=1}^n (\tilde{g}(S_i - s) - \max(S_i - s, 0)) + (1-\beta) \sum_{i=1}^n (\tilde{g}(s - S_i) - \max(s - S_i, 0)) + \frac{\mu}{2}(s - s_0)^2 \\
&\leq \frac{n\log 2}{\kappa} + \frac{\mu}{2}(s - s_0)^2,
\end{aligned}
\tag{27}
$$

and $\tilde{\rho}_\beta(s) - \rho_\beta(s) \geq \frac{\mu}{2}(s - s_0)^2$. By $\mu$-strong convexity and $\nabla\tilde{\rho}_\beta(\hat{s}^*) = 0$, we have

$$
\tilde{\rho}_\beta(s^*) - \tilde{\rho}_\beta(\hat{s}^*) \geq \frac{\mu}{2}(s^* - \hat{s}^*)^2. \tag{28}
$$

We next bound the term $\tilde{\rho}_\beta(s^*) - \tilde{\rho}_\beta(\hat{s}^*)$ as follows:

$$
\begin{aligned}
\tilde{\rho}_\beta(s^*) - \tilde{\rho}_\beta(\hat{s}^*) &= \tilde{\rho}_\beta(s^*) - \tilde{\rho}_\beta(\hat{s}^*) + \rho_\beta(s^*) - \rho_\beta(s^*) \\
&\leq \frac{n\log 2}{\kappa} + \frac{\mu}{2}(s^* - s_0)^2 + \rho_\beta(s^*) - \tilde{\rho}_\beta(\hat{s}^*) \\
&\overset{(a)}{\leq} \frac{n\log 2}{\kappa} + \frac{\mu}{2}(s^* - s_0)^2 + \rho_\beta(\hat{s}^*) - \tilde{\rho}_\beta(\hat{s}^*) \\
&\leq \frac{n\log 2}{\kappa} + \frac{\mu}{2}(s^* - s_0)^2 - \frac{\mu}{2}(\hat{s}^* - s_0)^2 \\
&\overset{(b)}{\leq} \frac{n\log 2}{\kappa} + \frac{\mu}{2}\epsilon_0^2,
\end{aligned}
\tag{29}
$$

where $(a)$ is due to $\rho_\beta(s^*) \leq \rho_\beta(\hat{s}^*)$ by the definition $s^*$, and $(b)$ is by Assumption 4.1. Finally, we have

$$
(s^* - \hat{s}^*)^2 \leq \frac{2}{\mu}\left(\tilde{\rho}_\beta(s^*) - \tilde{\rho}_\beta(\hat{s}^*)\right) \leq \frac{2n\log 2}{\mu\kappa} + \epsilon_0^2. \tag{30}
$$

This yields the desired result. $\qquad\square$

## A.2. Proof of Theorem 4.4

We have $|\bar{s}^{(T)} - s^*| \leq |\bar{s}^{(T)} - \hat{s}^*| + |\hat{s}^* - s^*| \leq \epsilon^{(T)} + \tilde{\epsilon}_0$, yielding

$$
\bar{s}^{(T)} - \epsilon^{(T)} - \tilde{\epsilon}_0 \leq s^* \leq \bar{s}^{(T)} + \epsilon^{(T)} + \tilde{\epsilon}_0. \tag{31}
$$

Then, for a data point $(X, Y)$ drawn from P in (1), we have

$$
\mathrm{P}\left(Y \in C^{\text{Q-DCP}}(X|\mathcal{D})\right) = \mathrm{P}\left(s(X,Y) \leq \bar{s}^{(T)} + \epsilon^{(T)} + \tilde{\epsilon}_0\right) \geq \mathrm{P}\left(s(X,Y) \leq s^*\right). \tag{32}
$$

Since $s^*$ is the empirical $(1-\alpha)(1+K/n)$-quantile in (4) and by [Theorem 4.3, Lu et al., 2023] with $w_k \propto n_k + 1$, we have $\mathrm{P}\left(s(X,Y) \leq s^*\right) \geq 1 - \alpha$, which implies the desired result. $\qquad\square$

## A.3. Proof of Theorem 5.2

Define the spectral gap $\varrho = 1 - |\lambda_2(\boldsymbol{W})|$. Then, the typical result of the consensus convergence in Xiao & Boyd (2004) yields the inequality

$$
\sum_{k\in\mathcal{V}} \|\boldsymbol{x}_k^{(t)} - \boldsymbol{p}\|^2 \leq (1 - \eta\varrho)^{2t} \sum_{k\in\mathcal{V}} \|\boldsymbol{x}_k^{(0)} - \boldsymbol{p}\|^2. \tag{33}
$$

In other words, the consensus error decays exponentially.

Let the histogram error as $\boldsymbol{e}_k^{(T)} = \boldsymbol{x}_k^{(T)} - \boldsymbol{x}$. To simplify the notation, we denote $m_{\alpha,k} = m_{\alpha,k}^{\text{H-DCP}}$, $\epsilon = \epsilon^{\text{H-DCP}}$, $\hat{S}_{i,k} = \Gamma(s(X_{i,k}, Y_{i,k}))$ as the $i$-th quantized score on device $k$, and $\hat{S}_{\text{test}} = \Gamma(s(X_{\text{test}, Y_{\text{test}}}))$ as the quantized score of the test point. By definition of $m_{\alpha,k}$, we have

$$
\begin{aligned}
m_{\alpha,k} &= \min\left\{ m \in [M] : \sum_{i=1}^{m} x_{i,k}^{(T)} \geq (1-\alpha)(1+\frac{K}{n}) + \epsilon \right\} \\
&= \min\left\{ m \in [M] : \sum_{i=1}^{m} \left( p_i + e_{i,k}^{(T)} \right) \geq (1-\alpha)(1+\frac{K}{n}) + \epsilon \right\} \\
&= \min\left\{ m \in [M] : \sum_{i=1}^{m} p_i \geq (1-\alpha)(1+\frac{K}{n}) + \epsilon - \sum_{i=1}^{m} e_{i,k}^{(T)} \right\},
\end{aligned}
\tag{34}
$$

Following the technique in (Lu et al., 2023), let $w_k = \frac{n_k+1}{n+K}$ and define the event E as

$$
\text{E} = \left\{ \forall k \in \mathcal{V}, \exists \pi_k, \left( \hat{S}_{\pi_k(1),k}, \hat{S}_{\pi_k(2),k}, \ldots, \hat{S}_{\pi_k(n_k+1),k} \right) = (s_{1,k}, s_{2,k}, \ldots, s_{n_k+1,k}) \right\},
\tag{35}
$$

where $\{s_{i,k}\}_{i \in [n_k+1], k \in \mathcal{V}}$ are the order statistics of the scores. For each device $k$, we also define $b_k(\gamma) = |\{\hat{S}_{i,k} \leq \gamma\}|$. By the definition of $m_{\alpha,k}$, $\frac{m_{\alpha,k}}{M}$ is the $\lceil (1-\alpha)(n+K) + n\epsilon - n\sum_{i=1}^{m_{\alpha,k}} e_{i,k}^{(T)} \rceil$-th smallest score in $\cup_{k=1}^{K}\{\hat{S}_{i,k}\}$ leading to

$$
\sum_{k=1}^{K} b_k\left( \frac{m_{\alpha,k}}{M} \right) = \left\lceil (1-\alpha)(n+K) + n\epsilon - n\sum_{i=1}^{m} e_{i,k}^{(T)} \right\rceil.
\tag{36}
$$

Then, we have

$$
\begin{aligned}
\text{P}\left( \hat{S}_{\text{test}} \leq \frac{m_{\alpha,k}}{M} \bigg| \text{E} \right) &= \sum_{k=1}^{K} w_k \cdot \mathbf{P}\left( \hat{S}_{\text{test}} \leq \frac{m_{\alpha,k}}{M} \bigg| \left\{ \hat{S}_{1,k}, \ldots, \hat{S}_{n_k,k}, S_{\text{test}} \right\} \text{ are exchangeable, E} \right) \\
&\geq \min\left\{ 1, \sum_{k=1}^{K} w_k \frac{b_k(\frac{m_{\alpha,k}}{M})}{n_k+1} \right\} \\
&= \min\left\{ 1, \frac{\left\lceil (1-\alpha)(n+K) + n\epsilon - n\sum_{i=1}^{m_{\alpha,k}} e_{i,k}^{(T)} \right\rceil}{n+K} \right\} \\
&\geq \min\left\{ 1, 1-\alpha + \frac{n\epsilon - n\sum_{i=1}^{m_{\alpha,k}} e_{i,k}^{(T)}}{n+K} \right\}.
\end{aligned}
\tag{37}
$$

We next bound the RHS. Using $\min\{x, y\} = y + \frac{x-y-|x-y|}{2}$, we have

$$
\begin{aligned}
\text{P}\left( \hat{S}_{\text{test}} \leq \frac{m_{\alpha,k}}{M} \bigg| \text{E} \right) &\geq 1 - \frac{1}{2}\alpha + \frac{1}{2}\frac{n\epsilon - n\sum_{i=1}^{m_{\alpha,k}} e_{i,k}^{(T)}}{n+K} - \frac{1}{2}\left| \frac{n\epsilon - n\sum_{i=1}^{m_{\alpha,k}} e_{i,k}^{(T)}}{n+K} - \alpha \right| \\
&= 1 - \frac{1}{2}\left( \alpha - \frac{n\epsilon}{n+K} \right) - \frac{1}{2}\frac{n\sum_{i=1}^{m_{\alpha,k}} e_{i,k}^{(T)}}{n+K} \\
&\quad - \frac{1}{2}\sqrt{\left( \alpha - \frac{n\epsilon}{n+K} \right)^2 + 2\left( \alpha - \frac{n\epsilon}{n+K} \right)\frac{n}{n+K}\left( \sum_{i=1}^{m_{\alpha,k}} e_{i,k}^{(T)} \right) + \frac{n^2}{(n+K)^2}\left( \sum_{i=1}^{m_{\alpha,k}} e_{i,k}^{(T)} \right)^2}.
\end{aligned}
\tag{38}
$$

Next, we bound the term $\left( \sum_{i=1}^{m_{\alpha,k}} e_{i,k}^{(T)} \right)^2$ as

$$\left( \sum_{i=1}^{m_{\alpha,k}} e_{i,k}^{(T)} \right)^2 \leq m_{\alpha,k} \sum_{i=1}^{m_{\alpha,k}} (e_{i,k}^{(T)})^2 \leq M \sum_{i=1}^{M} (e_{i,k}^{(T)})^2 \leq M \|\boldsymbol{e}_k^{(T)}\|^2, \tag{39}$$

where the first inequality follows from Jensen's inequality. This directly implies $\sum_{i=1}^{m_{\alpha,k}} e_{i,k}^{(T)} \leq |\sum_{i=1}^{m_{\alpha,k}} e_{i,k}^{(T)}| \leq \sqrt{M} \|\boldsymbol{e}_k^{(T)}\|$. Furthermore, by the convergence of consensus (33), we have

$$\|\boldsymbol{e}_k^{(T)}\|^2 \leq \sum_{k\in\mathcal{V}} \|\boldsymbol{e}_k^{(T)}\|^2 \leq (1-\eta\varrho)^{2T} \sum_{k\in\mathcal{V}} \|\boldsymbol{x}_k^{(0)} - \boldsymbol{p}\|^2 \leq K(1-\eta\varrho)^{2T} \max_{k\in\mathcal{V}} \|\boldsymbol{x}_k^{(0)} - \boldsymbol{p}\|^2. \tag{40}$$

The last term $\|\boldsymbol{x}_k^{(0)} - \boldsymbol{p}\|^2$ can be bounded as, for any $k \in \mathcal{V}$,

$$\|\boldsymbol{x}_k^{(0)} - \boldsymbol{p}\|^2 = \|\boldsymbol{x}_k - \frac{1}{K}\sum_{j=1}^{K} \boldsymbol{x}_j\|^2 \leq \frac{1}{K}\sum_{j=1}^{K} \|\boldsymbol{x}_k - \boldsymbol{x}_j\|^2 \leq \frac{2}{K}\sum_{j=1}^{K} \left( \|\boldsymbol{x}_k\|^2 + \|\boldsymbol{x}_j\|^2 \right). \tag{41}$$

To bound $\|\boldsymbol{x}_k\|^2$, we first recall that $\boldsymbol{x}_k = \frac{Kn_k}{n}\boldsymbol{p}_k$, where $\boldsymbol{p}_k = [p_{1,k}, \ldots, p_{M,k}]$ with $p_{m,k} = 1/n_k \sum_{i=1}^{n_k} \mathbb{I}\{\Gamma(S_{i,k}) = m/M\}$, $0 \leq p_{m,k} \leq 1$ and $\boldsymbol{1}^T\boldsymbol{p}_k = 1$. Then, we have

$$\|\boldsymbol{x}_k\|^2 = \frac{K^2n_k^2}{n^2}\|\boldsymbol{p}_k\|^2 = \frac{K^2n_k^2}{n^2}\sum_{m=1}^{M}(p_{m,k})^2 \leq \frac{K^2n_k^2}{n^2}\sum_{m=1}^{M}p_{m,k} = \frac{K^2n_k^2}{n^2}. \tag{42}$$

Thus,

$$\|\boldsymbol{e}_k^{(T)}\|^2 \leq \frac{2K^3(1-\eta\varrho)^{2T}}{n^2} \max_{k\in\mathcal{V}} \left\{ n_k^2 + \frac{1}{K}\sum_{j=1}^{K} n_j^2 \right\}. \tag{43}$$

To simplify the notation, we define the constant

$$\begin{aligned}
A &= \frac{n\sqrt{M}}{n+K} \sqrt{\frac{2K^3(1-\eta\varrho)^{2T}}{n^2} \max_{k\in\mathcal{V}} \left\{ n_k^2 + \frac{1}{K}\sum_{j=1}^{K} n_j^2 \right\}} \\
&= \frac{K\sqrt{2KM}(1-\eta\varrho)^T}{n+K} \sqrt{\max_{k\in\mathcal{V}} \left\{ n_k^2 + \frac{1}{K}\sum_{j=1}^{K} n_j^2 \right\}}.
\end{aligned} \tag{44}$$

Combining the above bounds, we have

$$\mathrm{P}\left( \hat{S}_{\text{test}} \leq \frac{m_{\alpha,k}}{M} \Big| \mathrm{E} \right) \geq 1 - \frac{1}{2}\left( \alpha - \frac{n\epsilon}{n+K} \right) - \frac{1}{2}A - \frac{1}{2}\sqrt{\left( \alpha - \frac{n\epsilon}{n+K} \right)^2 + 2A\left( \alpha - \frac{n\epsilon}{n+K} \right) + A^2}. \tag{45}$$

Let the RHS of (45) equal to $1 - \alpha$, yielding

$$\epsilon = \frac{n+K}{n}A. \tag{46}$$

Finally, the coverage guarantee of H-DCP follows

$$\mathrm{P}\left( Y_{\text{test}} \in \mathcal{C}_k^{\text{H-DCP}}(X_{\text{test}}|\mathcal{D}) \right) = \mathrm{P}\left( \hat{S}_{\text{test}} \leq \frac{m_{\alpha,k}}{M} \right) = \mathbb{E}_{\mathrm{E}}\left[ \mathrm{P}\left( \hat{S}_{\text{test}} \leq \frac{m_{\alpha,k}}{M} \Big| \mathrm{E} \right) \right] \geq 1 - \alpha. \tag{47}$$

# B. Additional Experimental Results

## B.1. Comparing with SOTA on a star topology

In the special case of a star topology, the communication cost of Q-DCP coincides with FCP, while H-DCP reduces to WFCP. Experimental results with $\alpha = 0.1$, comparing Q-DCP and H-DCP with FCP (Lu et al., 2023), FedCP-QQ (Humbert et al., 2023), and WFCP (Zhu et al., 2024), can be found in Table 1. These results show that the proposed protocols have comparable performance as the state of the art in terms of coverage and set size in the special case of a star topology. However, in contrast to existing schemes, H-DCP and Q-DCP apply to any network topology.

*Table 1.* Coverage and set size for Q-DCP, H-DCP, FCP, WFCP, and FedCP-QQ under CIFAR100 in a star topology with $\alpha = 0.1$.

|  | Q-DCP | H-DCP(WFCP) | FCP | FedCP-QQ |
|---|---|---|---|---|
| Coverage ($\pm$std) | $0.913 \pm 0.01$ | $0.92 \pm 0.014$ | $0.918 \pm 0.01$ | $0.996 \pm 0.003$ |
| Set Size ($\pm$std) | $11.35 \pm 0.89$ | $12.16 \pm 1.47$ | $11.98 \pm 0.99$ | $45.32 \pm 7.22$ |

## B.2. Convergence Results for Q-DCP and H-DCP

Setting $\epsilon_0 = 0.1$, Fig. 6 shows the coverage and set size versus the number of ADMM iterations $T$ for centralized CP and for Q-DCP on different graphs. We specifically consider the complete graph, torus, star, cycle, and chain, which are listed in order of decreasing connectivity. While coverage is guaranteed for any number of iterations $T$, Q-DCP is seen to require more rounds to achieve efficient prediction sets on graphs with weaker connectivity levels. Furthermore, Q-DCP achieves performance comparable to CP on all graphs at convergence.

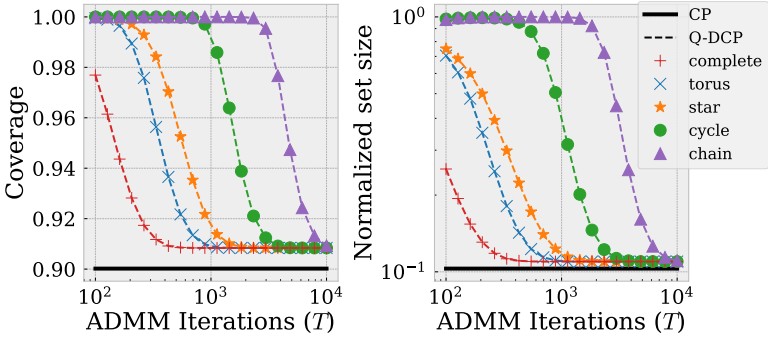

*Figure 6.* Coverage and normalized set size versus the number of ADMM iterations $T$ for CP and Q-DCP ($\alpha = 0.1$ and $\epsilon_0 = 0.1$).

Fig. 7 shows coverage and set size performance versus the number of consensus iterations $T$ for H-DCP on different graphs. Graphs with lower connectivity are observed to require more consensus iterations to give efficient prediction sets as in the case of Q-DCP presented in Fig. 6.

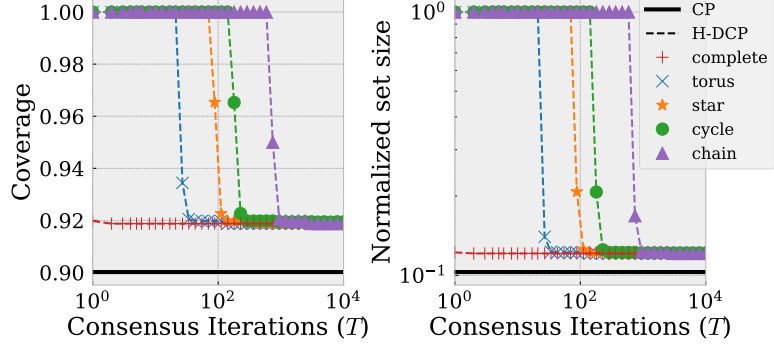

*Figure 7.* Coverage and normalized set size versus iterations $T$ for CP and H-DCP ($\alpha = 0.1$).

## B.3. Scalability and sensitivity of Q-DCP and H-DCP for network size

To validate the proposed method on a larger network, we considered a network with $K = 100$ devices, each of which includes data from a distinct class setting $T = 3000$ for both H-DCP and Q-DCP (with $\epsilon = 0.5$), experimental results can be found in Fig. 8 and Fig. 9. These results demonstrate that the proposed schemes are scalable to larger networks.

To evaluate the sensitivity of the performance of to the choice of $\epsilon_0$, we have evaluated coverage and set size for Q-DCP on Erdős–Rényi graphs with an increasing number of devices $K$, in which each edge is included in the graph with fixed probability 0.5. The 100 classes of CIFAR100 are divided (almost) equally among the $K$ devices (without replacement). Other parameters are the same as Sec. 6. For $\alpha = 0.1$ and $T = 3000$, the experimental result can be found in Fig. 10 with

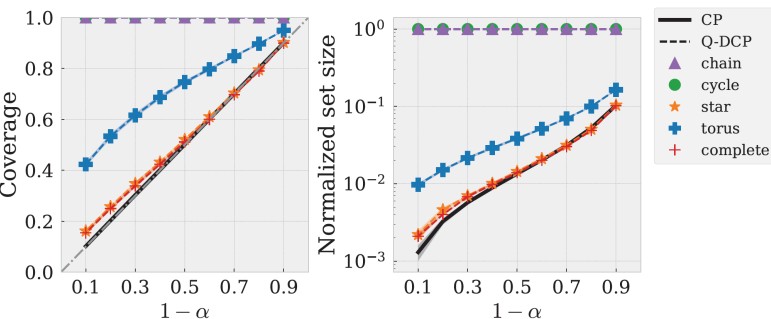

*Figure 8.* Coverage and normalized set size versus coverage level $1 - \alpha$ for CP and Q-DCP under CIFAR100 when Assumption 4.1 is satisfied ($K = 100$).

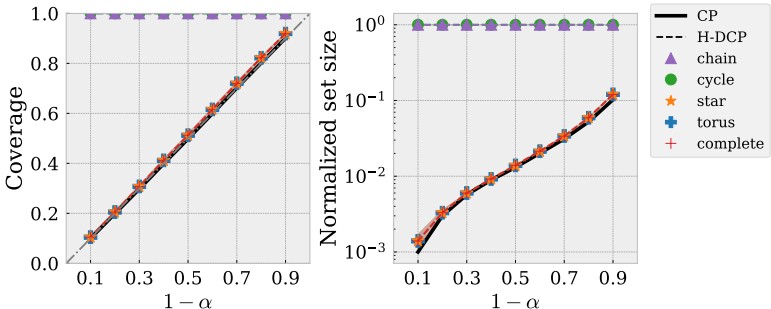

*Figure 9.* Coverage and normalized set size versus coverage level for CP and H-DCP under CIFAR100 ($K = 100$).

$\epsilon_0 = 1$ and in Fig. 11 with $\epsilon_0 = 0.1$. The level of connectivity increases with $K$ as the average spectral gap increases from 0.44 to 0.68. As a result, fixing $T$, the set size decrease as the level of connectivity increases. This observation is robust with respect to the choice of hyperparameter $\epsilon_0$. However, as verified by these results, the optimal choice of $\epsilon_0$ depends on the size of the network. In practice, for $\epsilon_0 = 1$, Assumption 4.1 is satisfied for all values of $K$ between 20 and 80, and thus Proposition 4.3 guarantees convergence to the target coverage probability $1 - \alpha = 0.9$ when $T$ is large enough. This is not the case for $\epsilon_0 = 0.1$, in which case Assumption 4.1 is violated as $K$ grows larger.

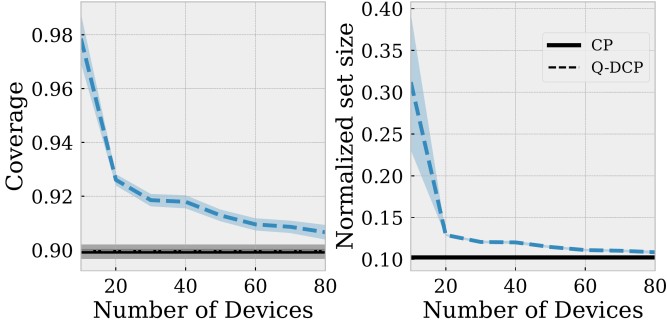

*Figure 10.* Coverage and normalized set size versus number of devices $K$ for CP and Q-DCP under CIFAR100 on Erdős–Rényi graph with connect probability 0.5 ($\alpha = 0.1, T = 3000, \epsilon = 1$).

### B.4. Ablation studies for Q-DCP

Fig. 12 to 14 show the effects of different choices of the smoothness parameter $\kappa$ and the regularization parameter $\mu$ on the coverage guarantee and set size for the torus graph, respectively. Fig. 12 shows that a small $\kappa$ will lead to an inefficient prediction set due to the inaccurate approximation of ReLU, while a large kappa will lead to a slow convergence speed of the ADMM yielding an inefficient prediction set as well. Fig. 13 demonstrates that a small $\mu$ causes a low ADMM convergence rate making the prediction set conservative. Fig. 14 shows that under the poor choices of $s_0$ and $\epsilon_0$ as in Fig. 3, in which Assumption 4.1 does not satisfied. One can observe that small $\mu$ gives the satisfaction of the coverage condition, while large

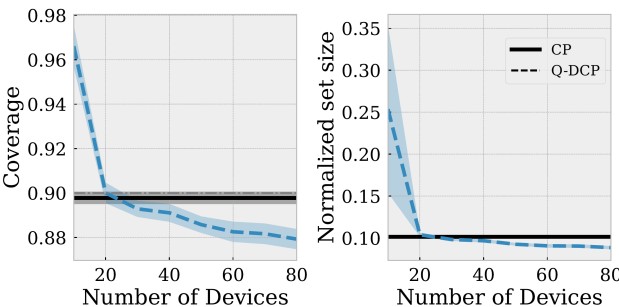

*Figure 11.* Coverage and normalized set size versus number of devices $K$ for CP and Q-DCP under CIFAR100 on Erdős–Rényi graph with connect probability 0.5 ($\alpha = 0.1, T = 3000, \epsilon = 0.1$).

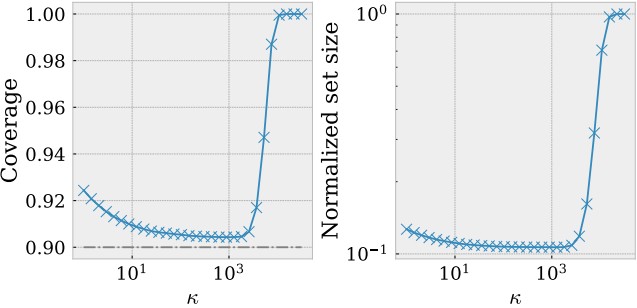

*Figure 12.* Coverage and set size versus $\kappa$ for Q-DCP. (Torus graph with $T = 1000, \alpha = 0.1, s_0 = \bar{s}$ and $\epsilon_0 = 0.1$)

$\mu$ leads to the solution of ADMM nearly same as $s_0$, yielding the violation of the coverage condition.

### B.5. Ablation study for H-DCP

Fig. 15 shows the coverage and set size versus quantization level $M$ for H-DCP on different graphs. One can observe that under limited consensus iterations, increasing the quantization level $M$ can make the estimated quantiles more accurate thus potentially providing more efficient prediction sets. On the flip side, it will lead to a larger error bound of the consensus and may violate the efficiency of the prediction on graphs with poor connectivity.

### B.6. Experimental results on PathMNIST

PathMNIST includes 9 classes and $107, 180$ data samples in total ($89, 996$ for training, $10, 004$ for validation, $7, 180$ for test). Medical datasets involve private and sensitive information, which are usually distributed over siloed medical centers, making a decentralized implementation practically relevant.

To this end, we trained a small-resnet14 as the shared model, and we considered a setting with $K = 8$ devices. Seven of the devices have data from only one class, while the last device stores data for the remaining two classes. For Q-DCP, the number of ADMM communication rounds was set as $T = 8000$, and the number of consensus iterations and the quantization level for H-DCP were set as $T = 80$ and $M = 100$. This way, both Q-DCP and H-DCP are subject to the same communication costs (in bits). Other settings remain the same as in Sec. 6. Results on coverage and normalized set size versus coverage level for Q-DCP and H-DCP, can be found in Fig. 16 and Fig. 17, respectively. These experimental results confirm the efficiency of the proposed methods for applications of interest.

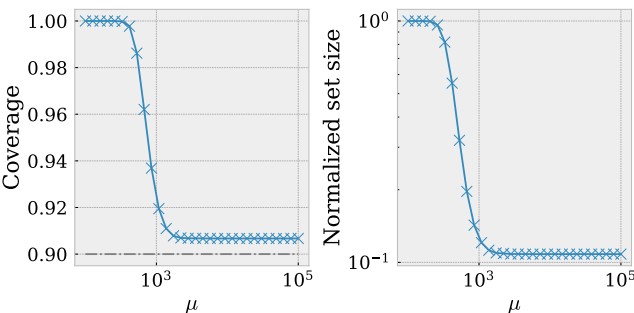

*Figure 13.* Coverage and set size versus $\mu$ for Q-DCP. (Torus graph with $T = 1000, \alpha = 0.1, s_0 = \bar{s}$ and $\epsilon_0 = 0.1$)

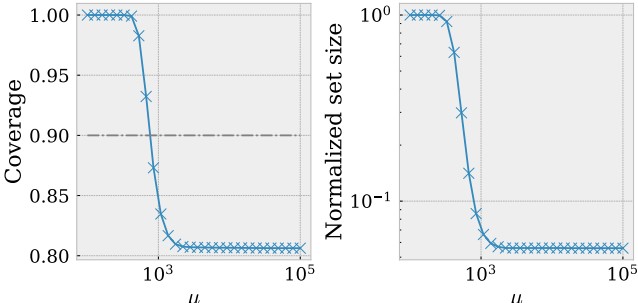

*Figure 14.* Coverage and set size versus $\mu$ for Q-DCP. (Torus graph with $T = 1000, \alpha = 0.1, s_0 = \min_{k \in \mathcal{V}} Q((1 - \alpha)(1 + K/n); \{S_{i,k}\}_{i=1}^{n_k})$ and $\epsilon_0 = 10^{-4}$)

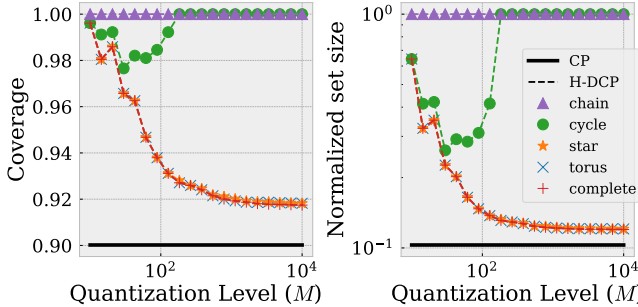

*Figure 15.* Coverage and set size versus quantization level $M$ for H-DCP. ($T = 150$ and $\alpha = 0.1$)

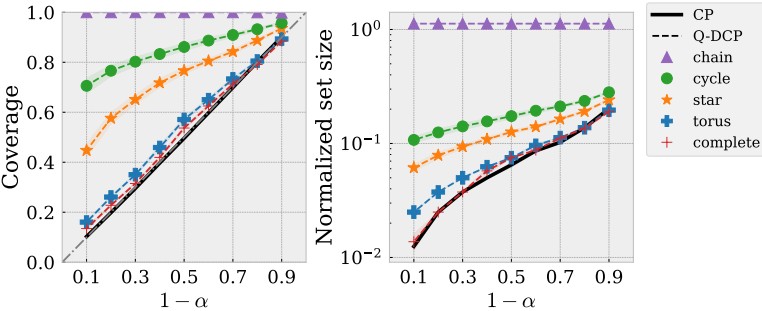

*Figure 16.* Coverage and normalized set size versus coverage level for CP and Q-DCP under PathMNIST.

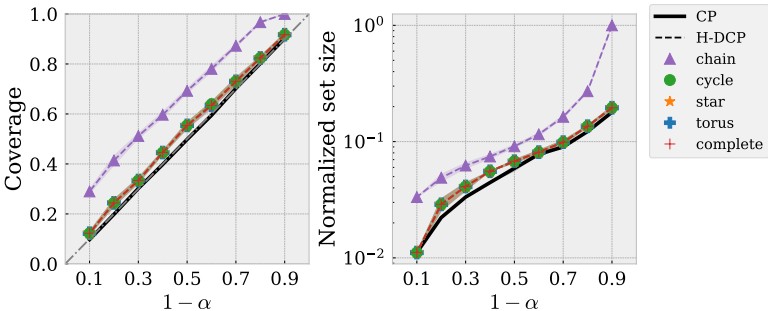

*Figure 17.* Coverage and normalized set size versus coverage level for CP and H-DCP under PathMNIST.

