# OpenReview forum: "Distributed Conformal Prediction via Message Passing"
_ICML.cc/2025/Conference — ICML 2025 poster_

### Official Review · Reviewer_iKT1 · 2025-03-09

**Overall Recommendation:** 4

**Summary:**

This work studies CP in a decentralised inference setting, where multiple devices share the same pre-trained model, and each device has a local calibration data set (motivated by e.g. privacy constraints). Given a common input, the devices aim at producing a prediction set that includes the true label of the test data with probability $1 − \alpha$. Two message-passing schemes are proposed for this Distributed CP (DCP) problem.

While the star graph topology was considered in prior works, this paper considers graphs where each node communicates only with its neighbours over an arbitrary graph topology.

-- update after rebuttal --

This is a well-written paper with solid theoretical contributions, so I maintain my evaluation. However, I continue to have reservations regarding the use of the terminology 'message passing' in the title. I would appreciate if the authors could address my questions listed under 'Other Strengths And Weaknesses' and 'Other Comments Or Suggestions' sections, not just those under the 'Questions For Authors' section.

**Claims And Evidence:**

This paper is of high quality in terms of presentation and novelty. The theoretical claims are explained clearly and supported by proofs. I've gone through the proofs for Q-DCP and they seem correct.

The experimental results are also comprehensive and compelling.

**Essential References Not Discussed:**

NA

**Experimental Designs Or Analyses:**

See above.

**Methods And Evaluation Criteria:**

Yes, the numerical experiments are rigorously designed. The proposed DCP methods are compared against the centralised CP as the benchmark, with graphs with different levels of connectivity.

**Other Comments Or Suggestions:**

- It might be beneficial to update the paragraph just above (23). Explain that the local vector $x_k$ represents the local estimate at device k of the global vector p. Add one sentence explaining why the update rule (23) is enforcing consensus among the devices and is therefore called the “linear consensus step”.

- line 346: nitpick: did you want to compare 1 real number vs M real numbers as the communication overhead? If so, it’s perhaps more accurate to say M-dim real vector instead of M-dim histogram vector.

**Other Strengths And Weaknesses:**

Title: A more precise and compelling title would better reflect the flexibility and novelty of the proposed DCP methods, and attract the intended readers. I’m not sure whether “message passing” is the best key word to use here because i) the main novelty is that arbitrary graph topologies are allowed instead of just simple star topologies, and ii) H-DCP doesn’t really use message passing?

**Questions For Authors:**

1. Do you require the calibration data to be iid across devices?

2. My understanding is that the constraint in (10) enforces the communication constraint based on the graph topology, is this correct? Could you elaborate?

3. line 175, right column: does “hyperparameter tuning” refer to tuning $\mu$ and $\kappa$?

**Relation To Broader Scientific Literature:**

This paper ties together quite a few interesting ideas with relevance extending beyond CP to optimisation theory, distributed computing, federated learning, and privacy.

A notable example is found in Q-DCP, where the authors transform the original objective (7) into a smooth, strongly-convex surrogate loss for accelerated convergence. This approach parallels the seminal adaptive regularisation technique from Bartlett, Hazan, and Rakhlin (2007), demonstrating how the paper builds upon established theoretical foundations while advancing them in new domains

**Theoretical Claims:**

I've gone through the proofs for Q-DCP and they seem correct. The proofs are well written.

---

> ### Author Rebuttal · Authors · 2025-04-01
>
> First, we are happy to consider your valuable suggestions in **Other Strengths And Weaknesses** and **Other Comments Or Suggestions**. For the other questions, please find the point-to-point response below.
>
> 1. > Do you require the calibration data to be iid across devices?
>
>    Please see our reply to comment 8 of Reviewer wc5C.
>
> 2. > My understanding is that the constraint in (10) enforces the communication constraint based on the graph topology, is this correct?
>
>    Yes, this is correct. In practice, the constraint in the problem (10) is equivalent to $s_i=z_{ij},s_j=z_{ij},$ for all $(i,j)\in \mathcal{E}$, where, $s_i$'s is the local copy of the shared optimization variable $s$ at device $i$, and $z_{ij}$ is an auxiliary variable imposing the desired consensus constraint between neighboring devices $i$ and $j$.
>
> 3. > line 175, right column: does hyperparameter tuning refer to tuning $\mu$ and $\kappa$?
>
>    Yes, this is correct.

---

> > ### Comment · Reviewer_iKT1 · 2025-04-01
> >
> > I acknowledge the authors' response.

---

### Official Review · Reviewer_wc5C · 2025-03-11

**Overall Recommendation:** 4

**Summary:**

This paper proposes a distributed way of achieving conformal prediction intervals. It extends the current literature by looking at graphs other than a star graph, and by looking at histogram summaries in addition to quantiles.

"## update after rebuttal: I have changed my recommendation to accept.

**Claims And Evidence:**

The theoretical claims are clear and the proofs are plausible.

**Essential References Not Discussed:**

None come to mind.

**Experimental Designs Or Analyses:**

The study is a comparison but perhaps more can be said. From Figures 2-4 it seems that all methods fail on the chain graph, in that they give prediction intervals which are far too large. What is the spectral gap for this graph? Is there an explanation? If one would want to apply any of the proposed algorithms, how could one check that they work well, are there guidelines?

**Methods And Evaluation Criteria:**

The benchmarking seems ok; a standard Cifar data set is used. It could be interesting to also look at synthetic examples.

**Other Comments Or Suggestions:**

It would be good to mention split conformal prediction already in the introduction.
It would be good to detail the choice of W already in 5.1.
What if the devices share both quantiles and histograms, could the method be improved?
If instead of sharing quantiles or histograms, the devices would share their CP intervals, would there be any mileage in that?

**Other Strengths And Weaknesses:**

The figures are very difficult to read; there are star and torus?

**Questions For Authors:**

Why do you need i.i.d. data for each device? Usually exchangeable scores suffice (an assumption which can again be weakened, see work by Rina Foygel Barber et al)

In the experiments how is the inefficiency determined when the distribution of the data is not available?

Why did you choose the torus graph for Figure 5 and not one of the other graphs?

**Relation To Broader Scientific Literature:**

There could have been a mention of federated learning more generally.

**Theoretical Claims:**

The proofs seem ok.

It is not clear how one can validate Assumption 4.1, that is, how one would find epsilon_0 which is non-trivial. The assumption seems to be key though.

---

> ### Author Rebuttal · Authors · 2025-04-01
>
> 1. > The benchmarking seems ok; a standard Cifar data set is used. It could be interesting to also look at synthetic examples.
>
>    Please refer to the rebuttal of Reviewer 6Gbu for details on experiments on a different data set.
>
> 2. > It is not clear how one can validate Assumption 4.1, that is, how one would find $\epsilon_0$ which is non-trivial. The assumption seems to be key though.
>
>    Indeed, as indicated in our paper, the difficulty in setting hyperparameter $\epsilon_0$ is one of the key motivations for introducing H-DCP, which does not require hyperparameter tuning. Please refer to page 6 lines 281-291(left) for further discussion on this point.
>
> 3. > The study is a comparison but perhaps more can be said. From Figures 2-4 it seems that all methods fail on the chain graph, in that they give prediction intervals which are far too large. What is the spectral gap for this graph? Is there an explanation? If one would want to apply any of the proposed algorithms, how could one check that they work well, are there guidelines?
>
>    The spectral gap of the chain graph is as low as $0.0123$, which is significantly smaller than other topologies. For example, the spectral gap of the star topology is $0.095$. As a result, as formalized by Proposition 4.3 and Theorem 5.2, the convergence in terms of communication rounds is decided by the spectral gap, which is thus slower for chain graphs.  It is indeed the results in Proposition 4.3 and Theorem 5.2 provide insightful guidelines about how various factors have an impact on the CP guarantee.
>
> 4. > There could have been a mention of federated learning more generally.
>
>    We would be happy to add further reference on federated learning.
>
> 5. > The figures are very difficult to read; there are star and torus?
>
>    We will increase the marker sizes of in the revised version.
>
> 6. > What if the devices share both quantiles and histograms, could the method be improved?
>
>    This is an interesting question. Combining advantages of both schemes, one could indeed apply H-DCP first to obtain an estimate of true global quantile, which is then used as prior information in the ADMM problem solved by Q-DCP. This can help mitigating the reliance of Q-DCP on the choice of hyperparameter $s_0$.
>
> 7. > If instead of sharing quantiles or histograms, the devices would share their CP intervals, would there be any mileage in that?
>
>    Sharing and merging their CP intervals is generally less efficient than computing a single set using all distributed data [R1, R2]. For example, supposing that we have $K$ prediction sets $\\{\mathcal{C}\_k\\}\_{k\in[K]}$ and merge them using majority vote as $\mathcal{C}^M:=\left\\{y\in\mathcal{Y}:1/K\sum\_{k=1}^K\mathbb{1}\\{y\in\mathcal{C}\_k\\}>\tau\right\\}$ for some $\tau \in [0,1)$. Then, by Theorem 2.1 in [R2], the majority vote procedure gives $1-\alpha/(1-\tau)$ coverage guarantee instead of $1-\alpha$ (see also Theorem 2.8 in [R2]).
>
>    [R1] M. Gasparin and A. Ramdas, “Conformal online model aggregation,” May 02, 2024, *arXiv*: arXiv:2403.15527.
>
>    [R2] Gasparin, M. and Ramdas, A. (2024). Merging uncertainty sets via majority vote. arXiv preprint arXiv:2401.09379.
>
> 8. > Why do you need i.i.d. data for each device? Usually exchangeable scores suffice...
>
>    We consider i.i.d. in the text to simplify the presentation. But indeed this assumption could be relaxed to exchangeability as done in [Assumption 4.1, Lu et al., 2023].
>
> 9. > In the experiments how is the inefficiency determined when the distribution of the data is not available?
>
>    We indeed do not have access to the distribution in the experiments. Therefore, the inefficiency defined as $\mathbb{E}|\mathcal{C}(X\_{\text{test}}|\mathcal{D})|$ is estimated using test data by $$1/|\mathcal{D}\_{\text{test}}|\sum\_{X\_{\text{test}}\in \mathcal{D}\_{\text{test}}}|\mathcal{C}(X\_{\text{test}}|\mathcal{D})|$$ in experiments.
>
> 10. > Why did you choose the torus graph for Figure 5 and not one of the other graphs?
>
>     This choice is motivated by the fact that the spectral gap of torus graph with $20$ devices is in a moderate regime, providing a balanced setting between a complete graph and a cycle graph in terms of the spectral gap, which significantly affects the prediction performance in line with Theorem 5.2.

---

> > ### Comment · Reviewer_wc5C · 2025-04-03
> >
> > Thank you for the explanations. Are you planning to revise the paper accordingly?

---

> > > ### Author Response · Authors · 2025-04-03
> > >
> > > Thank you. Yes, we will apply all the comments described in our reply.

---

### Official Review · Reviewer_7tEz · 2025-03-12

**Overall Recommendation:** 3

**Summary:**

The paper introduces two novel algorithms for distributed conformal prediction in decentralized networks. The Q-DCP employs ADMM to solve a distributed quantile regression problem with a smoothed pinball loss $ \tilde{\rho}\_\gamma(s)$ (incorporating a smoothing function $ \tilde{g}(x)$ and regularization term $ \frac{\mu}{2}(s-s\_0)^2$). After $T$ iterations, the device compute an average quantile estimate  $\bar{s}(T)$ and the authors derive an error bound $\epsilon\_{Q-DCP}$ such that $|\bar{s}(T) - s^*| \leq \epsilon\_{Q-DCP}$, ensuring the prediction set satisfies the coverage guarantee (Thm 4.4). The second one (or H-DCP) leverages a consensus-based histogram estimation approach. Each device quantizes its calibration scores into $M$ levels and exchanges these histogram vectors with its neighbors, allowing them to compute an average global histogram. From this, a quantile is estimated, and Theorem 5.2 guarantees that with error bound $\epsilon\_{H-DCP}$ linked to the consensus convergence, the resulting prediction set meets the desired coverage $P(Y \in C(X|D)) \geq 1-\alpha$.


***

 I appreciate the author's efforts to address my concerns. Based on the explanation and promised final revisions I increased my rating to 3

**Claims And Evidence:**

they are generally convincing

**Essential References Not Discussed:**

The application of message passing algorithms in optimization is not novel, such as prior work [1].

[1] Clarté, Lucas, and Lenka Zdeborová. "Building Conformal Prediction Intervals with Approximate Message Passing." arXiv:2410.16493

**Experimental Designs Or Analyses:**

-  the largest network tested has only 20 devices, which is very small for real-world decentralized applications

- The paper presents mean results but does not report confidence intervals or variance across multiple runs, which make it unclear whether observed differences are statistically significant or just noise. No comparison with other SOTA decentralized uncertainty quantification methods, such as federated conformal prediction or Bayesian approaches.

**Methods And Evaluation Criteria:**

While CIFAR-100 is a widely used benchmark, this paper does not test its methods in real-world distributed settings or more diverse datasets.

**Other Comments Or Suggestions:**

- I think integrating the pseudocode from the appendix into the main body could improve readability and help readers grasp the key ideas more effectively

- before equation (31), do you mean $\left|\bar{s}^{(T)}-\hat{s}^*\right|+\left|\hat{s}^*-s^*\right| \leq \epsilon^{(T)}+\tilde{\epsilon}_0$?

- equation (35), $E$ is already used

- equation (37),  the first line uses different probabilistic notation

**Other Strengths And Weaknesses:**

- While H-DCP somehow removes hyperparameter sensitivity, it suffers from a significantly higher communication cost per iteration due to the need for transmitting full histograms.

- The presentation of this paper fails to handle mathematical complexity effectively. While the problem formulation is reasonable to me, many of the introduced techniques appear somewhat ad hoc and lack a compelling motivation. Furthermore, the authors sometimes introduce notation without prior explanation, like $Z_i$ in (8), $E$ in line 230,231, etc

**Questions For Authors:**

- How does Q-DCP compare against recent federated conformal prediction approaches like [a,b] in terms of accuracy, communication cost, and robustness?

[a] Lu, Charles, et al. "Federated conformal predictors for distributed uncertainty quantification." International Conference on Machine Learning, 2023.

[b] Plassier, Vincent, et al. "Conformal prediction for federated uncertainty quantification under label shift." International Conference on Machine Learning, 2023.


- Does your Q-DCP’s hyperparameter sensitivity worsen as the e.g., network grows?

**Relation To Broader Scientific Literature:**

This work builds upon existing work in split conformal prediction, while also incorporating ideas from decentralized/federated optimization and message passing.

**Theoretical Claims:**

I didn't check the proofs line-by-line but they appear correct. The convergence error of ADMM, the bias from the smoothing approximation, etc, are just standard route in optimization analysis.

---

> ### Author Rebuttal · Authors · 2025-04-01
>
> 1. > The largest network tested has only 20 devices...
>
>    To validate the proposed method on a larger network, we considered a network with $100$ devices, each of which collects data from a distinct class, setting $T=3000$ for both H-DCP and Q-DCP (with $\epsilon=0.5$). The experiment results, which can be found [here](https://anonymous.4open.science/r/ICML_Rebuttal-AE74/Rebuttal_figures_and_tables.pdf) in Figures 3 and 4, demonstrate that the proposed schemes are scalable to larger networks.
>
> 2. > The paper presents mean results but does not report confidence intervals...
>
>    Thank you for suggesting adding error bar in the figures. You may find, e.g.,  Fig. 2 in the original draft, with an error bar of 95%-interval added [here](https://anonymous.4open.science/r/ICML_Rebuttal-AE74/Rebuttal_figures_and_tables.pdf) (See Figure 5).
>
> 3. > No comparison with other SOTA decentralized uncertainty quantification methods...
>    >
>    > How does Q-DCP compare against recent federated conformal prediction...
>
>    To the best of our knowledge, all previous distributed CP schemes applied only to the federated setting with a parameter server. Only when focusing on the special case of a star topology can we then compare the performance of the proposed protocols to the existing ones we are aware of, namely FedCP-QQ (Humbert et al., 2023), FCP (Lu et al., 2023) and WFCP (Zhu et al. (2024b)).
>
>    In this special case, the communication cost of Q-DCP coincides with FCP, while H-DCP reduces to WFCP. Experimental result with $\alpha=0.1$ can be found [here](https://anonymous.4open.science/r/ICML_Rebuttal-AE74/Rebuttal_figures_and_tables.pdf) in Table 1. These results, obtained at convenience, show that the proposed protocols have comparable performance to the existing state of the art in terms of coverage and set size in a star topology. However, in contrast to existing schemes, H-DCP and Q-DCP apply to arbitrary network topology.
>
> 4. > The application of message passing algorithms in optimization is not novel, such as prior work [1].
>
>    Please note that this interesting prior work [1] focuses on a fully centralized setting. This is fundamentally different from our decentralized setup. Accordingly, "message passing" in [1] refers to AMP, a Bayesian inference approach, while "message passing" in our work refers to gossip-style averaging consensus among distributed agents.
>
> 5. > The presentation of this paper fails to handle mathematical complexity effectively....
>    >
>    > the authors sometimes introduce notation without prior explanation...
>
>    We believe that our designs are formally well-motivated, and are validated by our theory.
>
>    - For Q-DCP: As discussed in line 194-202 (right), the smooth function $\tilde{g}(\cdot)$ and the regularization term in Eq. (8) aims for strong convexity, so as to ensure the linear convergence rate of ADMM, which has been theoretically verified in Proposition 4.3.
>
>    - For H-DCP: As discussed in lines 292-300 (left), the calibration score is quantized so as to support linear average consensus on the local histograms of the scores. As theoretically proved in Theorem 2, this ensures linear convergence and guarantees coverage.
>
>    The notation $Z_i$ in (8) was a typo, and it should be $S_i$. The notation $E$ refers to the number of edges $E=|\mathcal{E}|$.
>
> 6. > I think integrating the pseudocode...
>    >
>    > before equation (31), do you mean...
>
>    For the equation before Eq. (31), yes, you are correct. We would be happy to fit algorithm tables in the main text.
>
> 7. > Does your Q-DCP’s hyperparameter sensitivity worsen as the e.g., network grows?
>
>    The key hyperparameter to be selected in Q-DCP is $\epsilon_0$. To evaluate the sensitivity of the performance to the choice of $\epsilon_0$, we have evaluated Q-DCP on Erdős–Rényi graphs with an increasing number of devices $K$, in which each edge is included in the graph with a probability of 0.5. The 100 classes of CIFAR100 are uniformly at random (w/o replacement) divided among the $K$ devices. Other parameters are the same as the draft.
>
>    For $\alpha=0.1$ and $T=3000$, experimental results can found [here](https://anonymous.4open.science/r/ICML_Rebuttal-AE74/Rebuttal_figures_and_tables.pdf) in Figure 6 with $\epsilon_0=1$ and in Figure 7 with $\epsilon_0=0.1$. The average spectral gap increases with $K$ from 0.44 to 0.68. As a result, by fixing $T$, the set size decreases with the level of connectivity. This observation is robust against the choice of $\epsilon_0$. However, as verified by these results, the optimal choice of $\epsilon_0$ does depend on the size of network. In practice, for $\epsilon_0=1$, Assumption 4.1 is satisfied for all values of $K$ between $20$ and $80$, and thus convergence to the target coverage probability $1-\alpha=0.9$ is guaranteed when $T$ is large enough (see Proposition 4.3). This is not the case for $\epsilon_0=0.1$, when Assumption 4.1 is violated as $K$ grows larger.

---

### Official Review · Reviewer_6Gbu · 2025-03-15

**Overall Recommendation:** 3

**Summary:**

This paper addresses the challenge of conformal prediction in decentralized settings where multiple devices have limited calibration data and can only communicate with neighboring devices over arbitrary graph topologies. The authors propose two methods for distributed conformal prediction: Quantile-based Distributed Conformal Prediction (Q-DCP) and Histogram-based Distributed Conformal Prediction (H-DCP). Q-DCP employs distributed quantile regression enhanced with smoothing and regularization terms to accelerate convergence, solving the problem via ADMM. H-DCP uses a consensus-based histogram estimation approach, obtaining the global histogram of quantized calibration scores. Both methods provide theoretical coverage guarantees, with H-DCP offering hyperparameter-free guarantees at the cost of higher communication overhead.

**Claims And Evidence:**

The claims made in the paper are well-supported by theoretical analysis and empirical evidence. The theoretical results (Theorems 4.4 and 5.2) provide formal coverage guarantees for both proposed methods, with clear derivations and reasonable assumptions. The experimental results confirm these guarantees and illustrate the performance trade-offs across different network topologies, sample sizes, and hyperparameter settings. The comparison between Q-DCP and H-DCP regarding communication overhead versus hyperparameter sensitivity is particularly well-substantiated.

**Essential References Not Discussed:**

None

**Experimental Designs Or Analyses:**

The experimental design is comprehensive and appropriate. The authors evaluate their methods on Cifar100 data distributed across 20 devices in a non-i.i.d. manner, with each device assigned 5 unique classes. The evaluation covers various network topologies, hyperparameter settings, and communication budgets. The experiments verify theoretical results and provide practical insights on trade-offs between methods. The ablation studies effectively demonstrate the impact of key hyperparameters on performance.

**Methods And Evaluation Criteria:**

The proposed methods and evaluation criteria are appropriate for the problem. The authors use standard metrics in conformal prediction (coverage rate and prediction set size/inefficiency) to evaluate performance. The experimental setup using Cifar100 data distributed in a non-i.i.d. manner across devices is reasonable for demonstrating the methods' effectiveness. The comparison across different network topologies (chain, cycle, star, torus, and complete graph) provides insights into how connectivity affects performance.

**Other Comments Or Suggestions:**

None

**Other Strengths And Weaknesses:**

Strengths：
1. The paper presents a novel and theoretically sound approach to distributed conformal prediction that extends beyond the star topology assumed in prior work.
2. The authors provide strong theoretical guarantees with clear conditions under which they hold, along with empirical validation.
The comparative analysis between Q-DCP and H-DCP offers valuable insights into the trade-offs between communication efficiency and hyperparameter sensitivity.
3. The experimental evaluation is thorough, covering various network topologies, hyperparameter settings, and demonstrating convergence properties.


Weaknesses：
1. The experiments are limited to a single dataset (Cifar100). Including additional datasets, particularly those from domains mentioned as motivating applications (healthcare, IoT, autonomous vehicles), would strengthen the empirical evaluation.
2. The practical implementation details for large-scale distributed systems are somewhat limited. More discussion on handling device failures, communication delays, or asynchronous updates would enhance practicality.
3. While the methods address device-to-device communication, the initialization of both methods appears to require some coordination (e.g., for H-DCP, setting the consensus matrix W), which could be challenging in fully decentralized settings.
4. The study focuses on the post-hoc calibration of a shared pre-trained model, but does not explore scenarios where devices have different local models, which would be relevant for many real-world applications.

**Questions For Authors:**

1. How would the proposed methods perform if devices have heterogeneous computational capabilities or experience intermittent connectivity? Could the algorithms be adapted to handle asynchronous updates or device dropouts?
2. The current work assumes all devices share the same pre-trained model. How would the approaches need to be modified for scenarios where devices have different locally trained models, as might be the case in federated learning settings?
3. Could the Q-DCP approach be extended to provide localized or conditional coverage guarantees rather than just marginal coverage? This would be valuable for handling distribution shifts between devices.

**Relation To Broader Scientific Literature:**

The paper extends previous work on federated conformal prediction (e.g., FedCP-QQ, FCP, WFCP) which primarily addressed star topologies, to the more challenging case of arbitrary graph topologies. It integrates ideas from distributed optimization (ADMM), consensus algorithms, and conformal prediction. The work contributes to the growing literature on reliable and uncertainty-aware distributed machine learning, with connections to federated learning and distributed statistical estimation.

**Theoretical Claims:**

The theoretical claims in the paper appear sound. The paper provides detailed proofs for the main theorems (Theorems 4.4 and 5.2) in the appendix, establishing coverage guarantees for both Q-DCP and H-DCP. The proofs build on established results in distributed optimization and consensus algorithms, adapting them to the conformal prediction setting. The coverage guarantee for Q-DCP requires assumptions about parameter initialization that are carefully stated and verified in experiments.

---

> ### Author Rebuttal · Authors · 2025-04-01
>
> 1. > The experiments are limited to a single dataset (Cifar100). Including additional datasets...
>
>    Following your advice, we have evaluated the proposed scheme on a healthcare dataset, namely [PathMNIST](https://medmnist.com/). PathMNIST includes $9$ classes and 107,180 data samples in total.
>
>    We considered a setting with $K=8$ devices. Seven of the devices have data from only one class, while the last device stores data for the remaining two classes. For Q-DCP, we set $T=8000$, and for H-DCP, we set $T=80$ and $M=100$. This way, both Q-DCP and H-DCP are subject to the same communication costs (in bits). Other settings remain the same as in the paper. Experimental results, which can be found  [here](https://anonymous.4open.science/r/ICML_Rebuttal-AE74/Rebuttal_figures_and_tables.pdf) in Figures 1 and 2, confirm the efficiency of the proposed methods for applications of interest.
>
> 2. > Could the algorithms be adapted to handle asynchronous updates or device dropouts?
>
>    Leveraging existing literature on decentralized optimization and consensus, Q-DCP and H-DCP could indeed be extended to the above settings
>
>    - Q-DCP: Q-DCP is based on the ADMM protocol. An asynchronous version of ADMM was studied in [R1] for distributed convex optimization problems over a large-scale network with arbitrary topology. This approach may be applicable to the pinball loss minimization problem (7), yielding a generally slower convergence than with synchronous communications [Theorem 3.2, R1]. A setting with time-varying graphs, modeling communication outages, was studied in [R3] via first-order methods, which may also be applicable to problem (7).
>    - H-DCP: Asynchronous consensus algorithms with linear convergence rate were studied in [R2]. These may be leveraged to extend H-DCP to asynchronous settings. Furthermore, consensus protocols have also been widely studied for time-varying graphs [R4].
>
>    [R1] E. Wei and A. Ozdaglar, "On the $\mathcal{O}(1/k)$ Convergence of Asynchronous Distributed Alternating Direction Method of Multipliers," *2013 IEEE GlobalSIP*, 2013.
>
>    [R2] Y. Tian, Y. Sun and G. Scutari, "Achieving Linear Convergence in Distributed Asynchronous Multiagent Optimization," in *IEEE Trans. Autom. Control*., vol. 65, no. 12, pp. 5264-5279, Dec. 2020.
>
>    [R3] A. Nedić and A. Olshevsky, "Distributed Optimization Over Time-Varying Directed Graphs," in *IEEE Trans. Autom. Control*., vol. 60, no. 3, pp. 601-615, March 2015.
>
>    [R4] F. Xiao and L. Wang, “Asynchronous Consensus in Continuous-Time Multi-Agent Systems With Switching Topology and Time-Varying Delays,” *IEEE Trans. Autom. Control*., vol. 53, no. 8, pp. 1804–1816, Sep. 2008.
>
> 3. > the initialization of both methods appears to require some coordination...
>
>    In experiments, we choose the consensus matrix $\boldsymbol W$ in the standard form (See the first paragraph of Section 6.3). For this case, the eigenvalues of the Laplacian matrix $L$ can be obtained efficiently in a fully decentralized manner. See page 7 lines 340-348 (left) and reference [R5]. No other initialization settings for Q-DCP or H-DCP are required by coordination.
>
>    [R5] P. Di Lorenzo and S. Barbarossa, "Distributed Estimation and Control of Algebraic Connectivity Over Random Graphs," in *IEEE Trans. Signal Process*., vol. 62, no. 21, pp. 5615-5628, Nov.1, 2014.
>
> 4. > How would the approaches need to be modified for scenarios where devices have different locally trained models...
>
>    If the devices hold different local models, collaborative inference would have to be based on a different class of protocols. As an example, each device $k$ could first construct a local CP set $\mathcal{C}_k$ using the local model and data. Then, the CP sets could be aggregated via communications, which is a non-trivial problem that has been studied in [R6] and their later work using majority vote strategies. A fully decentralized implementation of these protocols in an arbitrary topology is an open problem. It is also important to note that these types of protocols would generally yield less efficient prediction sets when applied to settings in which agents share the same model [Theorem 2.8, R6].
>
>    [R6] Gasparin, Matteo, and Aaditya Ramdas. "Merging uncertainty sets via majority vote." *arXiv:2401.09379*, 2024.
>
> 5. > Could the Q-DCP approach be extended to provide localized or conditional coverage guarantees...
>
>    Following [R7], the localized coverage condition could be equivalently stated as Eq. (2.3) in [R7]. Accordingly, [R7] suggests approximating the localized coverage condition by solving the generalized optimization problem (2.4) in [R7]. It may be possible to generalize Q-DCP to address this problem, rather than problem (7) in our submission, in a decentralized way. Ensuring localized coverage opens up interesting research directions for future extensions of our work.
>
>    [R7] Gibbs I, Cherian JJ, Candès EJ. Conformal prediction with conditional guarantees. *arXiv:2305.12616*. 2023.

---

### Decision · Program_Chairs · 2025-05-01

**Decision:**

Accept (poster)

**Comment:**

This paper introduces two novel methods, Q-DCP and H-DCP, for distributed conformal prediction in decentralized settings with graph-based communication. The work is a significant extension of prior conformal prediction research, which focused primarily on star graphs, by addressing arbitrary graph topologies. The theoretical guarantees, including coverage bounds and convergence proofs, are rigorous and well-substantiated. The authors also provide experimental results to support their claims, exploring the trade-offs between the two methods in terms of communication cost, hyperparameter sensitivity, and efficiency under various graph structures.

The authors’ rebuttal meaningfully strengthens the submission. They addressed concerns about dataset limitations by adding experiments on PathMNIST, a healthcare dataset, and expanded the scale to 100 devices, demonstrating scalability and applicability to diverse real-world scenarios. They clarified key assumptions, such as the choice of the hyperparameter ε₀, and explained how H-DCP mitigates sensitivity to hyperparameter tuning. Additionally, they provided detailed insights into the impact of spectral gap on chain graph performance, offering a clear explanation for the observed inefficiencies.

While the paper still has some limitations, such as the lack of a direct comparison with other state-of-the-art methods in distributed uncertainty quantification and limited exploration of asynchronous communication or device heterogeneity, the rebuttal addresses these issues to a reasonable extent. The authors explained that no direct competitors exist for arbitrary graph topologies and highlighted how their methods subsume existing approaches in specific cases, such as star graphs. They also suggested feasible extensions to asynchronous settings and scenarios with locally trained models, providing valuable directions for future work.

In conclusion, this paper makes a substantial contribution to the field of distributed conformal prediction, offering strong theoretical and experimental support for its claims. The authors’ thoughtful responses and the additional experiments presented in the rebuttal further strengthen the paper. While some areas remain open for exploration, such as real-world deployment and performance on specific graph types, the overall quality and relevance of the work justify its acceptance.